# Monolayer Kagome metals $AV_3Sb_5$

Sun-Woo Kim [1,2,3,4], Hanbit Oh[2,4], Eun-Gook Moon [2] ✉ & Youngkuk Kim [1] ✉

Recently, layered kagome metals $AV_3Sb_5$ ($A$ = K, Rb, and Cs) have emerged as a fertile platform for exploring frustrated geometry, correlations, and topology. Here, using first-principles and mean-field calculations, we demonstrate that $AV_3Sb_5$ can crystallize in a mono-layered form, revealing a range of properties that render the system unique. Most importantly, the two-dimensional monolayer preserves intrinsically different symmetries from the three-dimensional layered bulk, enforced by stoichiometry. Consequently, the van Hove singularities, logarithmic divergences of the electronic density of states, are enriched, leading to a variety of competing instabilities such as doublets of charge density waves and $s$- and $d$-wave superconductivity. We show that the competition between orders can be fine-tuned in the monolayer via electron-filling of the van Hove singularities. Thus, our results suggest the monolayer kagome metal $AV_3Sb_5$ as a promising platform for designer quantum phases.

The kagome lattice refers to a two-dimensional (2D) planar crystal composed of corner-sharing triangles. Unique electronic structures emerge owing to the geometrical frustration of the lattice, featuring a flat band, a pair of Dirac points, and saddle-point van Hove singularities (VHSs). A prominent example of candidate kagome metals is the recently discovered vanadium-based kagome metals $AV_3Sb_5$ ($A$ = K, Rb, and Cs)[1,2]. A cascade of correlated electronic states have been observed in $AV_3Sb_5$, associated with charge density waves (CDWs)[2–17] and superconductivity[2,14–23]. These phases are reported to be accompanied by concomitant unexpected properties such as giant anomalous Hall effects[24,25] without long-ranged magnetic ordering[26], potential Majorana zero modes[14], and edge supercurrent[18]. The VHSs in conjugation with the Coulomb interaction is suggested as an impetus of the unconventional properties[27,28].

Although many outstanding materials have been found to comprise a kagome lattice in a layered form[29–31], the kagome lattice in genuine two dimensions is rare in nature. This scarcity leads to prior explorations of the kagome materials based on the assumption that a three-dimensional (3D) layered system can be regarded as decoupled kagome layers. Similarly, in the case of the vanadium-based kagome metals, current experiments are mainly focused on the 3D layered structures[1,2,2–5,11–16,18,19,21–26], while their theoretical analysis largely relies on an effective kagome model in two dimensions[6–9,20]. The dimensionality has been tacitly assumed as an irrelevant parameter, but this assumption has been generically refuted in layered systems[32,33]. A few research groups have made pioneering efforts to tackle this issue by successfully exfoliating thin films of $AV_3Sb_5$[34–36]. However, the importance of dimensionality in this family of kagome metals has remained elusive to date.

In this work, we theoretically demonstrate that the $AV_3Sb_5$ monolayer is different from the 3D layered bulk by performing density-functional theory (DFT) and mean-field theory (MFT) calculations. At the crux of our results is the absence of a dimensional crossover between the monolayer and the layered bulk. We argue that symmetry-lowering is inevitable in the monolayer, enforced by the stoichiometry of $AV_3Sb_5$. The reduced symmetries give rise to significant changes in the formation of VHSs. Notably, unconventional VHSs appear, referred to as type-II VHSs. As a consequence, enhanced electronic instabilities are observed, leading to the emergence of competing orders such as CDW doublets, time-reversal breaking CDWs, $s$- and $d$-wave super-conductivity. Our calculations predict that the $AV_3Sb_5$ monolayer can be thermodynamically stable. The stable $AV_3Sb_5$ monolayer becomes a unique platform to study the intriguing interplay between the VHSs and competing order parameters because any monolayer systems are under significantly enhanced fluctuations, as manifested in the celebrated Mermin-Wagner theorem[37–39]. In connection with future experiments, we also calculate the anomalous Hall conductivity that can probe the correlated orders. Possible experimental schemes are discussed to tune the electron-filling based on mechanical and chemical treatment.

[1]Department of Physics, Sungkyunkwan University, Suwon 16419, Republic of Korea. [2]Department of Physics, KAIST, Daejeon 34126, Republic of Korea. [3]Department of Materials Science and Metallurgy, University of Cambridge, 27 Charles Babbage Road, Cambridge CB3 0FS, UK. [4]These authors contributed equally: Sun-Woo Kim, Hanbit Oh. ✉e-mail: egmoon@kaist.ac.kr; youngkuk@skku.edu

## Results

### Crystal structure and symmetry

We begin by elucidating the similarities and differences between the crystal structures of the bulk and monolayer $AV_3Sb_5$ ($A$ = K, Rb, Cs). As delineated in Fig. 1a and b, both systems comprise multiple sub-layers. Most importantly, a 2D kagome sub-layer is formed from V atoms, coexisting with Sb sub-layers. While these are similar in both systems, differences arise from the alkali atoms $A$. In the monolayer (bulk) system, alkali metals energetically favor to form *rectangular (triangular)* sub-layers shown in Fig. 1b(a) (see Supplementary Note 1 for the detailed analysis of the energetics using first-principles calculations). The different formation of alkali atoms is traceable to the stoichiometry of $AV_3Sb_5$. The kagome layer of the monolayer takes all the electrons donated from the alkali atoms, while those in the bulk system are shared between the adjacent two sub-layers. Therefore, to preserve the stoichiometry, the number of neighboring alkali atoms is halved by doubling the unit cell, such that they form rectangular sub-layers.

The rectangular sub-layer with the doubled unit cell breaks translational and rotational symmetries of bulk $AV_3Sb_5$. The translational symmetry $\mathcal{T}_{1\times1}$ is reduced to $\mathcal{T}_{\sqrt{3}\times1}$, and correspondingly, a three-fold rotational symmetry $C_{3z}$ is lifted. This reduces the $D_{6h}$ symmetry of the bulk to $D_{2h}$ in the monolayer. We stress that the lowered symmetry of the monolayer $AV_3Sb_5$ is the fundamental symmetry, which can be stable even at room temperature protected by energy barriers from the stoichiometry enforcement, while the same symmetry is only reachable at low temperature from a cascade of phase transitions in bulk $AV_3Sb_5$[4,15]. We also note that any incommensurate CDW (IC-CDW) orders are prohibited at non-zero temperatures in monolayer systems unless either long-range interactions or substrate effects play a significant role. Thus, a dimensional crossover is

unlikely to occur from the bulk to the monolayer $AV_3Sb_5$, and unique properties arise as a result.

### Rearrangement of VHSs

Most importantly, the lowered symmetry of the monolayer $AV_3Sb_5$ leads to the rearrangement of the VHS in energy-momentum space. The mechanism of the rearrangement is illustrated in Fig. 1c, in which we trace the *k*-points that host the VHS. Hereafter, we refer to these momenta as the VHS points. The pristine BZ with the $\mathcal{T}_{1\times1}$ translational symmetry initially hosts three inequivalent VHS points at $M_i$ ($i$ = 1, 2, 3), as in the case of the bulk $AV_3Sb_5$ (left panel in Fig. 1c). Upon the zone folding by lowering $\mathcal{T}_{1\times1}$ to $\mathcal{T}_{\sqrt{3}\times1}$, the $M_1$ VHS point is folded to $\Gamma$ and the $M_1$ and $M_2$ VHS points are merged to the $M$ point of the reduced BZ (middle panel in Fig. 1c). The symmetry-lowering further hybridizes the two states at $M$ ($M_2$ and $M_3$), such that it annihilates the VHS at $M$ and creates four new VHS points off $M$, marked as $P_i$ ($i$ = 1, 2, 3, 4) in the right panel of Fig. 1c.

The rearrangement of VHS points is observed in our first-principles calculations. Figure 2a, b show exemplary DFT bands of $KV_3Sb_5$ with archetypal kagome bands distilled by our tight-binding theory (see Methods for the details of the TB model). In Fig. 2b, the divergence of the density of states (DOS) is clearly observed at $E = -6(+9)$ meV. A close inspection reveals that the diverging DOS at $E = -6$ meV arises from off high-symmetry momenta at $P_i = M + (\pm 0.054, \pm 0.021)$ Å$^{-1}$ (Fig. 2d), while the divergence at $E = 9$ meV arises from the exact high-symmetry $\Gamma$ point (Fig. 2c). In this respect, the VHS points at $E = 9$ meV ($E = -6$ meV) belong to the type-I (type-II) class, where the type-I (II) refers to a class of VHSs that originates from (off) time-reversal invariant momenta[40–42]. Our calculations further reveal that the type-II VHSs generically occur in the kagome

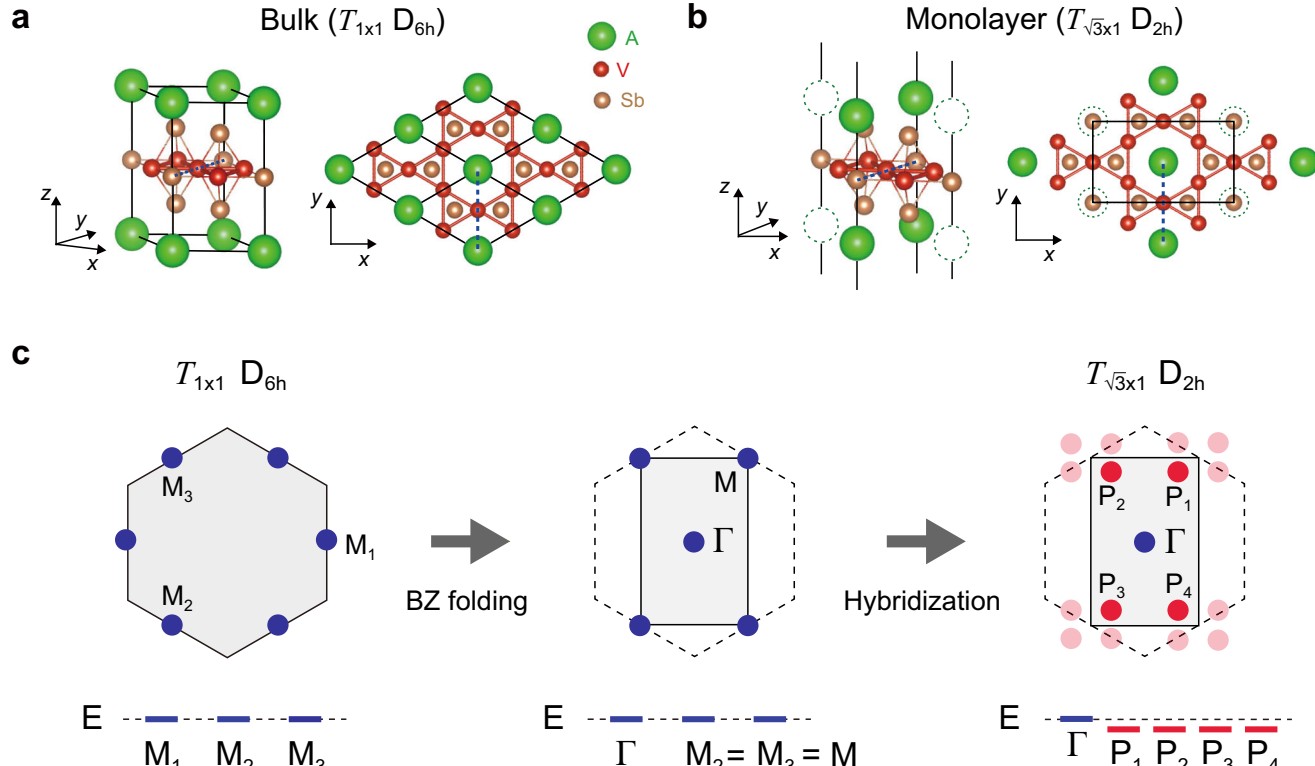

**Fig. 1 | Symmetry lowering and rearrangement of the van Hove singularity (VHS) in monolayer $AV_3Sb_5$. a, b** Atomic structures of the bulk and monolayer $AV_3Sb_5$ ($A$ = K, Rb, Cs). The monolayer structure preserves $\sqrt{3}\times1$ translational and $D_{2h}$ point group symmetries. The layered bulk preserves $1\times1$ translational and $D_{6h}$ point group symmetries. Black solid lines indicate the primitive unit cells and blue dashed lines indicate the *y*-axis. In **b**, dashed open circles represent vacant alkali

sites. **c** Schematic illustration of the VHS points in momentum space rearranged by symmetry lowering. The type-I (type-II) VHS points are marked by blue (red) circles. In the middle and right panels, the reduced (pristine) BZ of the $\sqrt{3}\times1$ ($1\times1$) unit cell is indicated by solid (dashed) line. The bottom panels delineate the energy level of the VHS points.

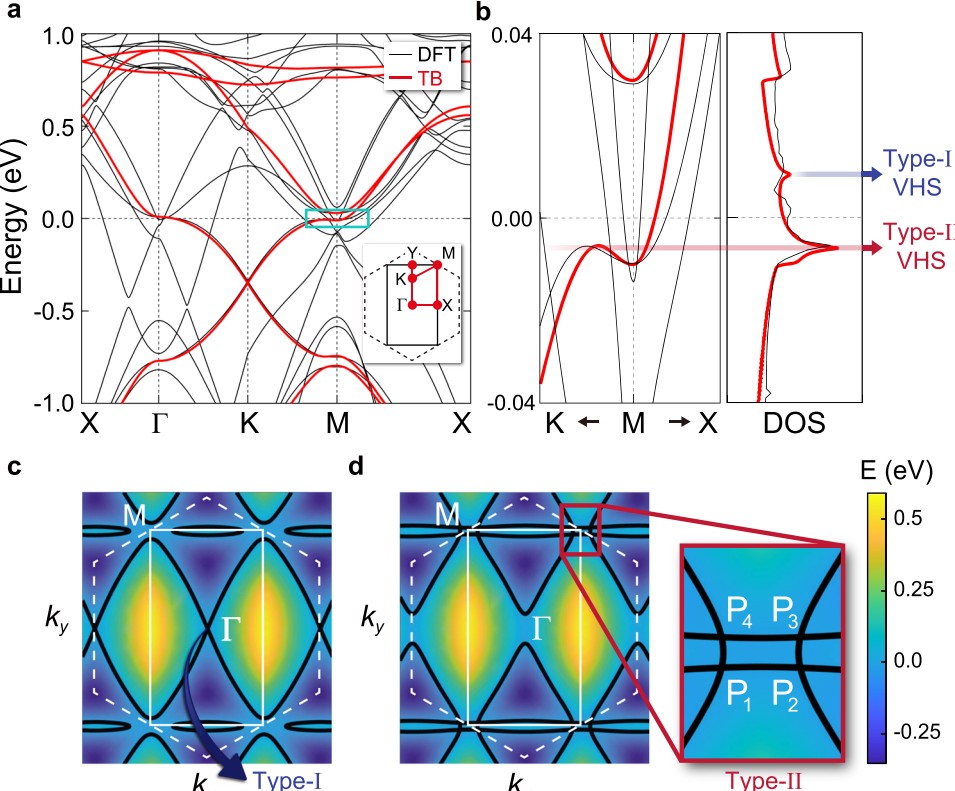

**Fig. 2 | VHSs in the monolayer $AV_3Sb_5$. a** DFT and TB band structures of the $KV_3Sb_5$ monolayer. The DFT (TB) bands are colored by black (red). The kagome bands are reproduced by using our TB model (see Methods for the details of the TB model). A solid rectangle (dashed hexagon) in the inset depicts the reduced (pristine) BZ of the $\sqrt{3} \times 1$ ($1 \times 1$) unit cell. **b** (Left panel) Magnified view of the blue box in **a**. (Right panel) Density of states (DOS) of $KV_3Sb_5$. The type-I and type-II VHSs are indicated by blue and red arrows, respectively. **c**, **d** Energy contours of the kagome bands calculated from the TB bands. Solid (dashed) white lines show the reduced (pristine) BZ. The contour curves in **c**, **d** correspond to $E = 9$ and $-6$ meV, respectively. The saddle points reside at the crossing points of the contour curves in **c**, **d**. The magnified view of the red box in **d** illustrates the position of type-II VHS points off $M$, marked by $P_i$ ($i = 1, 2, 3, 4$).

metals regardless of $A = K$, Rb, and Cs (see Supplementary Note 2 for the DFT results of the $RbV_3Sb_5$ and $CsV_3Sb_5$ monolayers).

The emergence of the type-II VHS is one of the key features of the monolayer $AV_3Sb_5$. Few remarks are as follows. First, the appearance of the type-II VHS off time-reversal invariant momenta is understood as a generic result attributed to the band hybridization allowed by the symmetry lowering. Since the two VHSs at $M_2$ and $M_3$ before the symmetry lowering are at the exact same energy level, they can be mixed together and contribute to generating the type-II VHSs (See Fig. 1c). On the other hand, the VHS at $M_1$ still remains a type-I VHS because no states are available to hybridize with. We find that the number of the generated type-II VHS points (in this case, four) is not universal and depends on the microscopic details of the system (see details in Supplementary Note 3.2). Second, the type-II VHS points $P_i$ ($i = 1, 2, 3, 4$) (Fig. 2d) consist of a mixed contribution from both the B and C sublattices (Supplementary Note 3, Fig. S5), referred to as a mixed-type flavor[12,20,43]. This is in contrast to the type-I VHS point at $\Gamma$ (Fig. 2c), which is purely contributed from the A sublattice, referred to as a pure-type flavor. Third, the increased number of VHS points quantitatively changes the characteristic of the diverging DOS at $-6$ meV. Namely, the peak at $-6$ meV is significantly enhanced than that of the type-I VHS at 9 meV (Fig. 2b), contributed from the quartet VHS points at $P_i$ ($i = 1, 2, 3, 4$). Such quantitative changes are of immediate impact on the electronic properties, such as instabilities driven by the type-II VHS, as we will show below.

**Instability of the monolayer**

The pristine monolayer harbors intrinsic instability captured in the electronic susceptibility. Figure 3a–d show the real part of the bare static charge susceptibility $\chi(\boldsymbol{q})$ at various fillings (see Methods). We find that the presence of both type-I and type-II VHSs leads to diversified instability sensitively depending on electron filling. For example, at the type-II VHS filling (Fig. 3a), a significant contribution arises from the $\boldsymbol{q}$ vectors that connect the van Hove saddle points. The peaks at $\boldsymbol{q}_1$ and $\boldsymbol{q}_2$ in $\chi(\boldsymbol{q})$ correspond to the nesting vectors mediating distinct $P_i$ points. The presence of these $\boldsymbol{q}$ vectors that are incommensurate to the reciprocal lattice vector necessitates the consideration of an incommensurate order parameter, such as IC-CDW phases. By contrast, when the chemical potential increases, the peaks quickly merge and give rise to the significant enhancement of $\chi(\boldsymbol{q})$ arises at the $M$ point (Fig. 3b–d). This significant peak at $\boldsymbol{q} = M$, as well as at $\boldsymbol{q} = \Gamma$ in the folded BZ, corresponds to the commensurate $2 \times 2$ CDW vectors. Notably, the large values of $\chi(\boldsymbol{q})$ at $M$ survive for a wide range of chemical potentials. These results suggest a significant role of $2 \times 2$ CDW orders in stabilizing the monolayer. Moreover, the sublattice character of $\chi(\boldsymbol{q})$ reveals that sublattice interference is significant in the monolayer (See Supplementary Note 4 for the calculations of sublattice-resolved susceptibility). This result warrants the consideration of longer-range interactions, which promotes unconventional orders as in the bulk cases[6,20,43,44].

The DFT phonon energy spectra also capture the instability of the pristine monolayer. We find that significant phonon softening arises at $M$ and $\Gamma$, associated with the displacement of the V atoms. Figure 3e shows a representative example of the DFT phonon spectra for monolayer $KV_3Sb_5$. The $\Gamma$ ($M$) point corresponds to the one (other two) of the $2 \times 2$ $3Q$ CDW vectors in the folded BZ. Thus, these negative branches are a clear indicative of the structural instability associated

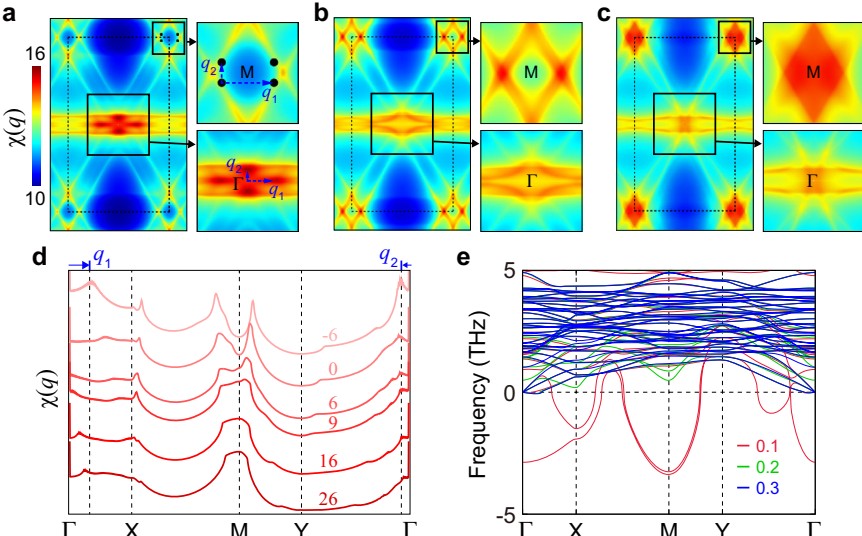

**Fig. 3 | Electronic susceptibility and phonon bands of the $A$V$_3$Sb$_5$ monolayer.** **a**–**c** Electronic susceptibility $\chi(\boldsymbol{q})$ is calculated in 2D $\boldsymbol{q}$-space for **a** the type-II VHS filling ($\mu = -6$ meV), **b** neutral filling ($\mu = 0$ meV), and **c** type-I VHS filling ($\mu = 9$ meV). The right panels show a magnified view of the boxed areas of the left panel near the high-symmetric Γ and $M$ points, respectively. $\boldsymbol{q}_1$ and $\boldsymbol{q}_2$ in **a** represent the nesting vectors that connect the type-II VHS points (black circles). **d** Line profiles of $\chi(\boldsymbol{q})$ along the high-symmetry lines of the BZ for various chemical potentials. The corresponding chemical potential (in meV) is shown in each profile. The momenta correspond to $\boldsymbol{q}_1$ and $\boldsymbol{q}_2$ vectors in **a** are indicated by arrows with vertical lines. **e** Phonon bands of the pristine KV$_3$Sb$_5$ monolayer as a function of electronic temperature. The electronic temperature effect is introduced by tuning the smearing factor $\sigma$ (from 0.1 to 0.3 eV) of the Fermi-Dirac distribution function.

with the $2 \times 2$ distortion of V atoms, similar to the bulk counterparts[17,45–47]. We also find that these $2 \times 2$ CDW instabilities are generically present in monolayer $A$V$_3$Sb$_5$ for $A$ = K, Rb, and Cs (see Supplementary Fig. S3). We note that the monolayer hosts the broadened softened phonon modes in the momentum space compared to the bulk[17], which signals various CDW instabilities other than $2 \times 2$. Nonetheless, we confirm that the $2 \times 2$ CDW instability is a leading instability by performing energy profile analysis for the various imaginary phonon modes (see Supplementary Note 5.1). Thus, it is imperative to consider the $2 \times 2$ $3Q$ CDWs in the monolayer, such as the star of David (SD) and inverse star of David (ISD) distortions in the monolayer.

To investigate the role of the electronic instability in $2 \times 2$ CDW instabilities, we additionally calculated the phonon spectra as a function of electronic temperature, smearing factor $\sigma$, as shown in Fig. 3e. The softened phonon at $M$ is lifted as we destroy the Fermi surface by increasing the electronic temperature. These calculations demonstrate that the electronic instability associated with the VHS plays a role in the formation of structural distortion. Since the DFT phonon spectra include the effects of electron-phonon coupling, our results suggest that either electronic or phononic contributions cannot be excluded in forming structural instability. In addition, the occurrence of the $2 \times 2$ instability in the phonon spectra is plausible in view of the electronic susceptibility calculations. Our DFT calculations are performed off the type-II VHS filling, in which large susceptibility values arise near the $M$ point. While the reduced symmetry featured in the monolayer leads to the rearrangement of the VHSs into type-I and type-II VHSs at different fillings and correspondingly two types of distinct CDW orders—$2 \times 2$-commensurate CDWs and IC-CDWs, we believe that the commensurate orders are more likely to occur in the monolayer, especially at non-zero temperature. The pristine monolayer $A$V$_3$Sb$_5$ is under strong fluctuations as dictated in the celebrated Mermin-Wagner theorem, prohibiting any IC-CDW orders at non-zero temperature. Thus, hereafter we mainly focus on commensurate order parameters in this work and discuss the possibility of incommensurate orders at zero temperature in Supplementary Note 11.

## Competing orders

The rearranged VHSs manifest their properties in competing orders of correlated electronic states. In monolayer $A$V$_3$Sb$_5$, we employ the standard mean-field theory with the constructed TB model and uncover phase diagrams with CDWs and superconductivity (SC). The onsite and nearest-neighbor Coulomb interactions are introduced,

$$H_{\mathrm{int}} = U \sum_{\boldsymbol{R}} \sum_{\alpha} n_{\boldsymbol{R},\alpha\uparrow} n_{\boldsymbol{R},\alpha\downarrow} + V \sum_{\boldsymbol{R}} \sum_{\substack{(\alpha,\beta) \\ \sigma,\sigma'}} n_{\boldsymbol{R},\alpha,\sigma} n_{\boldsymbol{R},\beta,\sigma'}, \tag{1}$$

where $U$ ($V$) describes the on-site (nearest-neighbor) density-density type interaction and $\boldsymbol{R}$, $\alpha$, and $\sigma$ represent the lattice site, sublattice, and spin, respectively. We consider two classes of order parameters, CDWs and SC, which can significantly reduce the energy by gapping out the Fermi surface with diverging DOSs. CDWs and SC are of particular interest as they have been observed in the bulk $A$V$_3$Sb$_5$ in a variety of forms, including SD, ISD, and time-reversal symmetry breaking (TRSB) CDWs[6,7,10,17].

Remarkably, any CDW order in the monolayer $A$V$_3$Sb$_5$ forms a doublet, as illustrated in Fig. 4c. The doublet formation of CDWs is one of the key characteristics of the monolayer $A$V$_3$Sb$_5$, originating from the reduced symmetry. The lowered $\mathscr{T}_{\sqrt{3}\times 1}$-translational symmetry of $A$V$_3$Sb$_5$ monolayer plays a crucial role to double the CDW channels of the higher $\mathscr{T}_{1\times 1}$-symmetry, and the number of multiplets is solely determined by their quotient group, $\mathscr{T}_{1\times 1}/\mathscr{T}_{\sqrt{3}\times 1} = \mathbb{Z}_2$. For example, the two SD-CDW phases, dubbed SD-1 and SD-2, are the $\mathbb{Z}_2$ members, distinguished by the alkali chains hosted on and off the center of SD, respectively, as illustrated in Fig. 4c.

The corresponding phenomenological Landau theory of the doublet CDWs becomes exotic. Introducing a bosonic real two-component spinor, $\Psi_{\mathrm{SD}}^T \equiv (\psi_{\mathrm{SD}-1}, \psi_{\mathrm{SD}-2})$ with order parameters of SD-1 ($\psi_{\mathrm{SD}-1}$) and SD-2 ($\psi_{\mathrm{SD}-2}$), the Landau functional for the SD-CDW phases is given by

$$\mathscr{F}_L(\Psi_{\mathrm{SD}}) = r_{\mathrm{SD}}\left(\Psi_{\mathrm{SD}}^\dagger \Psi_{\mathrm{SD}}\right) + s_{\mathrm{SD}}\left(\Psi_{\mathrm{SD}}^\dagger \hat{\rho}_z \Psi_{\mathrm{SD}}\right) + \cdots, \tag{2}$$

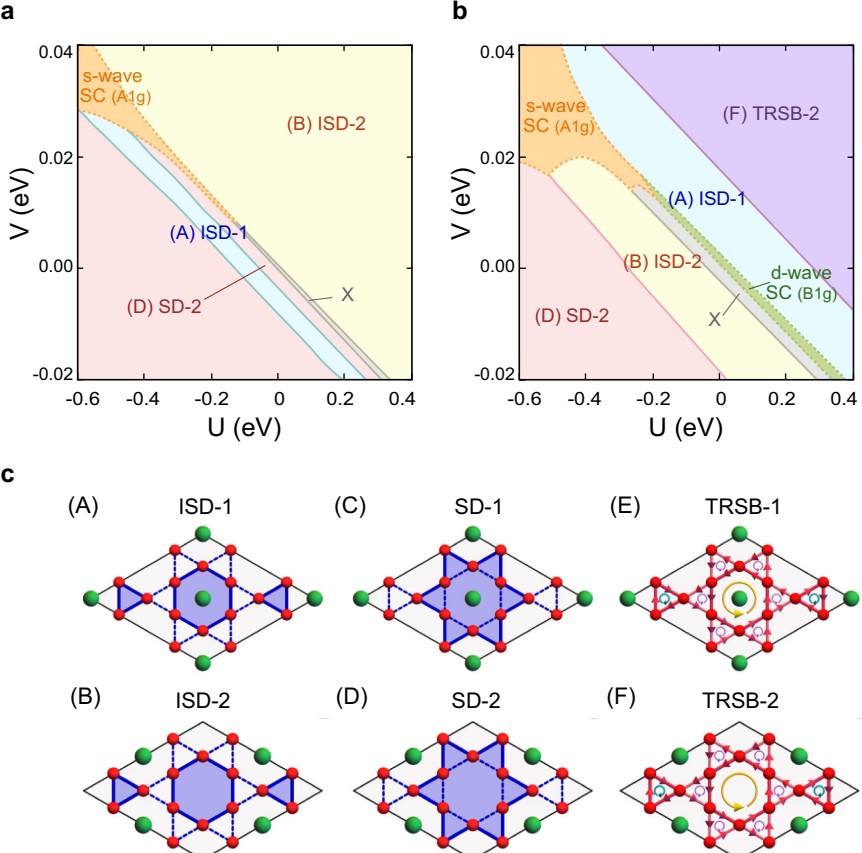

**Fig. 4 | Phase diagrams at $T \to 0^+$ and corresponding electronic orders in the $A V_3 Sb_5$ monolayer. a, b** Phase diagrams at the chemical potentials **a** $\mu_1 = -6$ meV and **b** $\mu_2 = 35$ meV in $U$-$V$ space, where $U$ and $V$ are the onsite and the nearest neighbor density-density type interactions. Different CDW and SC orders are indicated by different colors. The gap functions of s-wave and d-wave superconducting states correspond to $A_{1g}^{(1)}$ and $B_{1g}^{(1)}$ provided in Table 2. The superconducting phases only exist at zero temperatures, and their phase boundaries are denoted by dashed lines to contrast the boundaries of commensurate CDWs denoted by the solid lines. $X$ denotes a phase unidentified within our mean-field scheme. **c** Schematic illustration of six distinct CDW configurations. (c1-c2) Inverse star of David (ISD)-1,2, (c3-c4) star of David (SD)-1,2, (c5-c6) time-reversal symmetry breaking (TRSB)-1,2. In (c1-c4), solid and dashed blue lines indicate distinct bonding strengths. In (c5-c6), red arrows on the bonds indicate the direction of current bond orders. The oriented circles represent magnetic fluxes threading the hexagons and triangles where their magnitudes are proportional to the radius (see Supplementary Note 6 for the detailed descriptions). The corresponding CDW and SC order parameters are detailed in Tables 1 and 2.

with phenomenological constants $r_{SD}$ and $s_{SD}$. Here, the Pauli matrix $\hat{\rho}_z$ describes the spinor space, and higher order terms are omitted for simplicity. The $s_{SD}$-term describes a free-energy difference between SD-1 and SD-2, which is nonzero when the $\mathscr{T}_{1\times1}$-translational symmetry is broken. Depending on $s_{SD}$, the system energetically favors one of the doublet CDWs, enriching phase diagrams of the monolayer $A V_3 Sb_5$.

Our mean-field analysis indeed finds enriched phase diagrams of the $A V_3 Sb_5$ monolayer. We consider six configurations of CDWs (see Fig. 4c) and nine spin-singlet channels of SCs (see Table 2). In Fig. 4a, b, we illustrate representative mean-field phase diagrams of KV₃Sb₅ in $U$-$V$ space obtained at two different chemical potentials $\mu_1 = -6$ meV and $\mu_2 = 35$ meV, where the zero chemical potential is set to the neutral filling. We emphasize that our mean-field analysis of $A V_3 Sb_5$ monolayer is reliable at zero and very low temperatures since any monolayer systems suffer from significant thermal fluctuations. Thus, the phase diagrams of Fig. 4 need to be understood as the ones in the limit of lowering temperature down to zero, $T \to 0^+$. In what follows, we point out key observations made from the phase diagrams.

First, five distinct CDW orders can be accessible by fine-tuning the chemical potential $\mu$. For example, in the vicinity of type-II VHS at $\mu_1 = -6$ meV (Fig. 4a), SD-2 and ISD-2 dominantly occur with sizable regions of ISD-1 in the energy ranges of $-0.6$ eV $< U < 0.4$ eV and $-20$ meV $< V < 40$ meV. Similarly, a TRSB-2 CDW phase is uncovered under the condition $\mu \geq 30$ meV (Fig. 4b) in a wide range of $U$ and $V$

values. Moreover, the SD-1 phase appears near the type-I VHS at $\mu_3 = 9$ meV (see Supplementary Note 9). A small variation of chemical potential $-6$ meV $< \mu < 40$ meV can tune the types and flavors of the VHSs, which should enable an on-demand onset of a variety of CDW phases, ranging from ISD-1/2, SD-1/2, to TRSB-2.

Second, competition between CDWs and SC is generically observed. A conventional $s$-wave SC phase is observed near $\mu_1 = -6$ meV, which competes with ISD-1/2 and SD-2 at negative $U$ and positive $V$ as shown in Fig. 4a, b. Similarly, an unconventional $d$-wave SC phase is observed near $\mu_2 = 35$ meV. This competes with ISD-1 at positive $U$ and negative $V$ interactions as shown in Fig. 4b. Note that the chemical potential for the $d$-wave SC is quite higher than the energy of the type-II VHS $\mu_1$ and closer to the edge of the high energy band, which indicates that the VHS itself is not the unique reason to stabilize SC or CDW phases. The interplay between the VHS, filling, and interaction strengths should be considered together.

The final observation from our mean-field study is the nontrivial topology of the correlated CDW gaps. Notably, a non-zero Chern number $\mathscr{C}$ is induced in the energy spectra when gapped by the two time-reversal symmetry broken CDW phases TRSB-1 and TRSB-2. For example, the lowest unoccupied and highest occupied energy spectra of TRSB-1 (TRSB-2) host the Chern number $\mathscr{C} = -2$ and $\mathscr{C} = -1$ ($\mathscr{C} = -3$ and $\mathscr{C} = 2$), respectively. As shown in Fig. 5a, the different Chern numbers between the two phases arise due to the concurrent

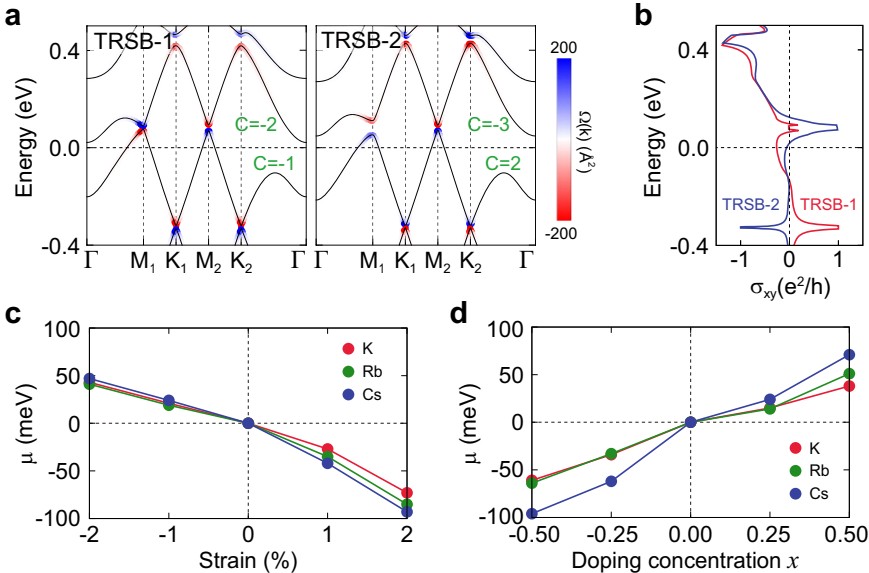

**Fig. 5 | Topological charge density waves and engineering of chemical potential. a** TB band structures of the TRSB-1 and TRSB-2 CDW phases. The Berry curvature $\Omega_z(\boldsymbol{k})$ are overlaid at the corresponding $\boldsymbol{k}$-points in the bands. The Chern numbers $\mathscr{C}$ induced in the bands are shown in green color. **b** Anomalous Hall conductivity $\sigma_{xy}$ as a function of energy $E$ for the TRSB-1 and TRSB-2 CDW phases. **c** DFT calculations of the chemical potential $\mu$ as a function of an uniform biaxial strain in $A$V$_3$Sb$_5$ for $A$ = K, Rb, and Cs. **d** DFT calculations of the chemical potential $\mu$ as a function of doping concentration $x$ for $A_{1+x}$V$_3$Sb$_5$. The detailed DFT band structures under the biaxial strain and alkali doping are provided in Supplementary Note 12.

sign-change of the Berry curvature at high-symmetry momenta $M_1$, $K_1$, and $K_2$. We note that the monolayer in the time-reversal symmetry broken CDW phase hosts the Fermi pockets that carry the Berry curvature, referred to as the Fermi Chern pockets[48] (see Supplementary Fig. S13). However, unlike the bulk case[48], asymmetric distribution of the Berry curvature occurs near the $M_1$ and $M_2$ in the monolayer due to the symmetry lowering. This results in distinct experimental observables, such as anomalous Hall conductivity $\sigma_{xy}$ (see detailed comparison between bulk and monolayer in Supplementary Note 6). In Fig. 5b, we calculate $\sigma_{xy}$ of the TRSB phases in the monolayer, which demonstrates the enriched structure of the anomalous Hall conductivities, distinct between TRSB-1 and TRSB-2. A sign change of $\sigma_{xy}$ is also found as a function of chemical potential near the Fermi energy $E = 0$. We believe this nontrivial behavior featured in $\sigma_{xy}$ can be readily observed in the Hall current measurements, leading to the experimental discovery of the TRSB-CDW phases.

The exotic orders should be accessible in monolayer $A$V$_3$Sb$_5$ in a controlled fashion. Our mean-field phase diagrams (Fig. 4a, b) show that the occurrence of a specific electronic order is highly contingent upon the correct filling of electrons, which can be fine-tuned via mechanical and chemical means. For example, by applying a uniform biaxial strain in a range of ± 2% variations, $\mu$ can be tuned from 50 meV to −100 meV (Fig. 5c). We also find that $U$ and $V$ values are functions of applied strains by performing the ab initio calculations with constrained random-phase approximation[49]. This indicates that different phases in the mean-field phase diagrams are accessible using strains (See Supplementary Note 10 for the detailed discussion and computational methods). Moreover, the light doping of alkali atoms $A$ (see Fig. 5d) or substituting Sb with Sn[50] can be a fine knob to adjust the chemical potential. Owing to the 2D geometry of the $A$V$_3$Sb$_5$ monolayer, we believe that there exist more opportunities (such as ionic gating) to tailor the competing orders hosted therein.

## Discussion

We have so far investigated unique features of the $A$V$_3$Sb$_5$ monolayer. Our system is unlike the bulk, hosted in a distinct symmetry class. The contrast is even more evident in the phase diagrams that we calculated

with the bulk $D_{6h}$ and the monolayer $D_{2h}$ symmetries, respectively (see Supplementary Fig. S19 for the $D_{6h}$ phase diagrams). The lowered $D_{2h}$ symmetry in the monolayer features a tendency to foster the CDW orders. This is in line with the previous experiments, in which a CDW order is observed to suppress SC as the thickness of the $A$V$_3$Sb$_5$ film decreases[34,35].

Our study arguably suggests that monolayer $A$V$_3$Sb$_5$ should be an exciting platform for studying intriguing 2D physics, such as the interplay between thermally suppressed IC-CDW orders and the type-II VHSs. At zero temperature, an IC-CDW associated with the type-II VHSs appears in the ($U \neq 0$, $V = 0$) phase space. We investigate the two different fillings ($\mu = -5$, $-7$ meV) near type-II VHSs and find that the IC-CDW phase is accessible with a negative $U$ in the fine-tuned range of chemical potential (see Supplementary Fig. S21). Thus, not only the electron filling but also interaction strengths are important to stabilize the IC-CDW phases. At any non-zero temperature, the quasi-long range IC-CDW order can exist, and uncovering the possibility of topological phase transitions such as Kostelitz-Thouless transitions is certainly an important issue in both theoretical and experimental regards. Therefore, we believe that the $A$V$_3$Sb$_5$ monolayer offers timely new physics in the kagome metal community, calling for future sophisticated theoretical and experimental studies.

Our results also provide further insights into the unconventional CDW orders of the bulk systems. Notably, our DFT phonon and electronic susceptibility calculations have shown that the 2 × 2 instability of the bulk systems remains a robust feature of the V-based kagome metals against the weakening of the VHS nesting by symmetry-lowering. We have attributed its rationale to significant electronic instability remaining at $\boldsymbol{q} = M$ for a wide range of electron filling. Similar mechanisms may explain the robustness of the CDW orders in the bulk systems under a weakened nesting condition due to, for example, out of the exact VHS filling[51]. In this regard, further studies in the monolayer should be complementary in pinning down the origin of the CDW orders. The observation of CDW without acoustic phonon anomaly and evolution of CDW amplitude modes may be important to resolve the CDW mechanism of the monolayer as discussed in the bulk case[52,53].

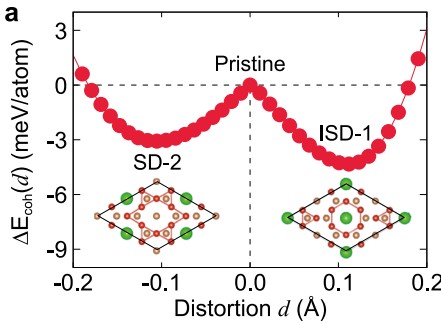

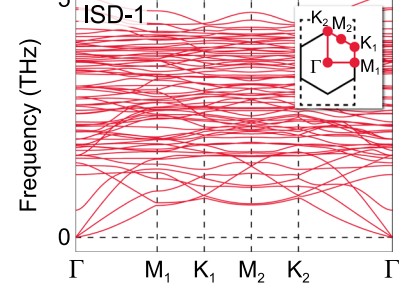

**Fig. 6 | Stability of the $AV_3Sb_5$ monolayer. a** Cohesive energy difference (per atom) as a function of the CDW distortion $d$ of the $KV_3Sb_5$ monolayer. The cohesive energy difference is defined as $\Delta E_{coh}(d) = E_{coh}(d=0) - E_{coh}(d)$, where the positive (negative) value of the distortion $d$ represents the shift of the V atom in the ISD-1 (SD-2) phase with respect to the pristine structure. **b** Phonon bands of monolayer $KV_3Sb_5$ in the ISD-1 phase. In the inset, the solid and dashed lines represent the BZs for the $2 \times 2$ and $\sqrt{3} \times 1$ structures, respectively. For Rb and Cs, see Supplementary Fig. S4.

We conclude our discussion by arguing that the $AV_3Sb_5$ monolayer should be possible to synthesize based on the following facts. First, the cohesive energy generically indicates the thermodynamic stability of the monolayer systems. The cohesive energy is calculated as 3.8 eV/atom for all three alkali atoms (see details in Supplementary Note 1). This value is comparable to the bulk value of ~ 3.9 eV/atom, supporting the thermodynamical stability of the monolayer. In Fig. 6a, we plot the cohesive energy difference of the ISD-1 and SD-2 phases $\Delta E_{coh}(d) = E_{coh}(d=0) - E_{coh}(d)$ as a function of the distortion of V atoms $d$. The CDW phases form a local minimum in the configuration space with an energy barrier of around 4 meV/atom. Second, our DFT phonon calculations indicate that the monolayer should be dynamically stable. Exemplary phonon bands of the ISD-1 phase are shown in Fig. 6b, which are clean of imaginary frequencies, showing the dynamical stability of the monolayer structure. Finally, the exfoliation energies of the monolayer are calculated as 42, 45, and 45 meV/$Å^2$ for $A$ = K, Rb, and Cs, respectively. These values are amount to existing two-dimensional materials, such as graphene (~ 21 meV/$Å^2$)[54], hBN (~ 28 meV/$Å^2$)[54], and $Ca_2N$ (~ 68 meV/$Å^2$)[55]. We note that the recent experiments have successfully exfoliated thin layers of $AV_3Sb_5$ up to five layers using the taping methods[35]. Current developments of the chemical solution reaction method could be an appropriate technique to weaken the interlayer interaction of the bulk system and separate the monolayer[56].

## Methods

### Tight-binding model

We construct a TB model for the monolayer with $D_{2h}$ symmetry in the $\sqrt{3} \times 1$ unit cell. Introducing the six-component spinor, $\Psi_{\boldsymbol{k}}^T = (A_{\boldsymbol{k}}^T, B_{\boldsymbol{k}}^T, C_{\boldsymbol{k}}^T)$ with $\alpha_{\boldsymbol{k}}^T = (\alpha_{1,\boldsymbol{k}}, \alpha_{2,\boldsymbol{k}})$, the Hamiltonian becomes $H_0 = \sum_{\boldsymbol{k}} \Psi_{\boldsymbol{k}}^\dagger \mathscr{H}_0(\boldsymbol{k}) \Psi_{\boldsymbol{k}}$. The indices for sublattice $\alpha \in \{A, B, C\}$ and site $i \in \{1, 2\}$ are used. The Bloch Hamiltonian $\mathscr{H}_0(\boldsymbol{k})$ is given by

$$
\begin{aligned}
\mathscr{H}_0(\boldsymbol{k}) = & -t[(c_1\lambda_1 - s_1\lambda_4 + c_2\lambda_2 - s_2\lambda_5 + 2c_3\lambda_3) \otimes I_2 \\
& + (c_1\lambda_1 + s_1\lambda_4 + c_2\lambda_2 + s_2\lambda_5) \otimes \sigma_x] \\
& -t_2[(c_4\lambda_1 - s_4\lambda_4 + c_5\lambda_2 - s_5\lambda_5) \otimes I_2 \\
& + (c_4\lambda_1 + s_4\lambda_4 + c_5\lambda_2 + s_5\lambda_5 + 2c_6\lambda_3) \otimes \sigma_x] \\
& + \epsilon I_6 + \frac{\delta\epsilon}{6}[(2\lambda_0 + 3\lambda_7 + \sqrt{3}\lambda_8) \otimes \sigma_z] - 2\delta t s_3[\lambda_6 \otimes \sigma_z],
\end{aligned} \quad (3)
$$

where $(c_i, s_i) \equiv (\cos \boldsymbol{k} \cdot \boldsymbol{r}_i, \sin \boldsymbol{k} \cdot \boldsymbol{r}_i)$ with $\boldsymbol{r}_1 = \frac{1}{2}(\sqrt{3}, 1)$, $\boldsymbol{r}_2 = \frac{1}{2}(\sqrt{3}, -1)$, $\boldsymbol{r}_3 = (0, 1)$, $\boldsymbol{r}_4 = \boldsymbol{r}_2 - \boldsymbol{r}_3$, $\boldsymbol{r}_5 = \boldsymbol{r}_1 + \boldsymbol{r}_3$, and $\boldsymbol{r}_6 = \boldsymbol{r}_1 + \boldsymbol{r}_2$. Here, $t$, $t_2$, $\epsilon$, $\sigma_i$ and $\lambda_i$ are nearest-neighbor hopping, next nearest-neighbor hopping, onsite energy, $2 \times 2$ Pauli matrices and $3 \times 3$ Gell-Mann matrices, respectively. The sublattice information and the definition of the Gell-Mann matrices are provided in Supplementary Fig. S5a and Supplementary Note 7, respectively. The last two terms give rise to onsite energy

difference $2\delta\epsilon$ between two A sublattices and staggered hopping $\pm \delta t$ between the B and C sublattices, respectively. These two terms lead to the symmetry-lowering from $\mathscr{T}_{1 \times 1} D_{6h}$ to $\mathscr{T}_{\sqrt{3} \times 1} D_{2h}$. With parameters $(\epsilon, t, t_2, \delta\epsilon, \delta t) = (0.01, 0.42, 0.03, -0.033, 0.01)$, our TB model well reproduces the DFT bands as well as the irreducible representations of the VHS bands at $\Gamma$ and $M$ ($A_g$) for $KV_3Sb_5$ (see Fig. 2a, b). For Rb and Cs, we also find good agreement between the TB and DFT (see Supplementary Note 2). The TB band structures of the TRSB-1/2 phases in Fig. 5a are obtained by using parameters $(\epsilon, t, t_2, \delta\epsilon, \delta t, \Phi) = (0.035, 0.42, 0.03, -0.033, 0.01, 0.07)$ where $\Phi$ is a CDW magnitude of the TRSB-1/2 phases (see Supplementary Note 8 for detailed order parameters of the TRSB-1/2 phases).

### Electronic susceptibility

We calculate the real part of the bare electronic susceptibility in the constant matrix approximation $\chi(\boldsymbol{q})$ given by

$$
\chi(\boldsymbol{q}) \equiv -\sum_{\boldsymbol{k}, \mu, \nu} \left[ \frac{n_F(E_\mu(\boldsymbol{k})) - n_F(E_\nu(\boldsymbol{k} + \boldsymbol{q}))}{E_\mu(\boldsymbol{k}) - E_\nu(\boldsymbol{k} + \boldsymbol{q})} \right], \quad (4)
$$

where $n_F(\epsilon)$ is the Fermi-Dirac distribution and $(\mu, \nu)$ is band index. For the calculation of $\chi(\boldsymbol{q})$ in Fig. 3, TB bands with $200 \times 348$ $\boldsymbol{k}$-points and $\beta = 1000$ are used.

### Mean-field theory

We perform the conventional zero-temperature mean-field method to investigate the interaction effects. Six distinct CDW configurations and nine spin-singlet SC orders are chosen to uncover CDW and SC instabilities in our analysis (see Fig. 4c and Table 1, 2). We introduce the CDW and SC order parameters $(\Phi, \Delta)$ that break translational and U(1)

**Table 1 | CDW orders in the mean-field analysis**

| Label | Order parameter ($\Phi_{AB}, \Phi_{AC}, \Phi_{BC}$) | Pattern | Subgroup |
|---|---|---|---|
| ISD-1 | $\Phi(1, 1, 1)$, $\Phi < 0$ | (c1) | $D_{2h}$ |
| ISD-2 | $\Phi(-1, 1, -1)$, $\Phi < 0$ | (c2) | |
| SD-1 | $\Phi(1, 1, 1)$, $\Phi > 0$ | (c3) | $D_{2h}$ |
| SD-2 | $\Phi(-1, 1, -1)$, $\Phi > 0$ | (c4) | |
| TRSB-1 | $\Phi(i, -i, i)$, $\Phi \neq 0$ | (c5) | $C_{2h}$ |
| TRSB-2 | $\Phi(-i, -i, -i)$, $\Phi \neq 0$ | (c6) | |

Both time-reversal symmetric and asymmetric types of CDWs are considered. Translational symmetry-lowering of monolayer $AV_3Sb_5$ diversifies the order parameters, leading to a doublet of CDWs (CDW-1/2). When time-reversal symmetry is preserved, the sign of $\Phi$ specifies the different CDW patterns resulting in the ISD and SD depending on $\Phi < 0$ and $\Phi > 0$, respectively. Without time-reversal symmetry, the TRSBs with opposite signs of $\Phi$ are equally classified. The real space patterns of the CDW orders are illustrated in Fig. 4c. Here, the $4 \times 4$ bond matrices are introduced with $O_1(\boldsymbol{k}) = (-\phi_1 I_2 + \phi_1^* \sigma_x) \otimes \sigma_z$, $O_2(\boldsymbol{k}) = \sigma_z \otimes (-\phi_2 I_2 + \phi_2^* \sigma_x)$, $O_3(\boldsymbol{k}) = \phi_3 \sigma_y \otimes \sigma_y + \phi_3^* \sigma_z \otimes \sigma_z$ and $\phi_i \equiv \exp(\boldsymbol{k} \cdot \boldsymbol{r}_i)$ (See Eq. (5)).

**Table 2 | SC orders in the mean-field analysis**

| R | Pairing in 1×1 unit cell | Pairing in 2×2 unit cell $\hat{\Gamma}_R(\boldsymbol{k})$ | Label | Basis function |
|---|---|---|---|---|
| $A_g$ | $\frac{1}{\sqrt{3}}\lambda_0, \frac{1}{\sqrt{2}}\left[\frac{\sqrt{3}}{2}\lambda_7 + \frac{1}{2}\lambda_8\right],$ | $\frac{1}{\sqrt{3}}L_0^0, \frac{1}{\sqrt{2}}\left[\frac{\sqrt{3}}{2}L_7^0 + \frac{1}{2}L_8^0\right],$ | $\Gamma_{A_g}^{(1)}, \Gamma_{A_g}^{(2)},$ | $x^2, y^2, z^2$ |
| | $\frac{1}{\sqrt{2}}\left[c_1\lambda_1 + c_2\lambda_2\right],$ | $\frac{1}{\sqrt{2}}\left[c_1L_1^+ + c_2L_2^+ - (s_1L_4^- + s_2L_5^-)\right],$ | $\Gamma_{A_g}^{(3)},$ | |
| | $c_3\lambda_3$ | $\left[c_3L_3^+ - s_3L_6^-\right]$ | $\Gamma_{A_g}^{(4)}$ | |
| $B_{1g}$ | $\frac{1}{\sqrt{2}}\left[-\frac{1}{2}\lambda_7 + \frac{\sqrt{3}}{2}\lambda_8\right],$ | $\frac{1}{\sqrt{2}}\left[-\frac{1}{2}L_7^0 + \frac{\sqrt{3}}{2}L_8^0\right],$ | $\Gamma_{B_{1g}}^{(1)},$ | $xy$ |
| | $\frac{1}{\sqrt{2}}\left[c_1\lambda_1 - c_2\lambda_2\right]$ | $\frac{1}{\sqrt{2}}\left[c_1L_1^+ - c_2L_2^+ - (s_1L_4^- - s_2L_5^-)\right]$ | $\Gamma_{B_{1g}}^{(2)}$ | |
| $B_{2u}$ | $\frac{1}{\sqrt{2}}\left[s_1\lambda_4 - s_2\lambda_5\right]$ | $\frac{1}{\sqrt{2}}\left[s_1L_4^+ - s_2L_5^+ - (c_1L_1^- - c_2L_2^-)\right]$ | $\Gamma_{B_{2u}}^{(1)}$ | $y$ |
| $B_{3u}$ | $\frac{1}{\sqrt{2}}\left[s_1\lambda_4 + s_2\lambda_5\right],$ | $\frac{1}{\sqrt{2}}\left[s_1L_4^+ + s_2L_5^+ + (c_1L_1^- + c_2L_2^-)\right],$ | $\Gamma_{B_{3u}}^{(1)},$ | $x$ |
| | $s_3\lambda_6$ | $\left[s_3L_6^+ - c_3L_3^-\right]$ | $\Gamma_{B_{3u}}^{(2)}$ | |

Spin-singlet superconducting pairings are classified by irreducible representations (R) of the point group $D_{2h}$.
The condensation of $\hat{\Gamma}_R$ channels can lower the interaction energy, which preserves the $\mathscr{T}_{1\times1}$-translational symmetry. The momentum dependence of pairing gap functions is abbreviated by $(c_i, s_i) \equiv (\cos \boldsymbol{k} \cdot \boldsymbol{R}_i, \sin \boldsymbol{k} \cdot \boldsymbol{R}_i)$. To express pairing functions in both 1×1 and 2×2 unit cells, the 3×3 Gell Mann matrices $\lambda_a$ and the 12×12 matrices $L_a^{\pm,0} = \lambda_a \otimes M_a^{\pm,0}$ are introduced with 4×4 matrices $M_a^{\pm,0}$ (see Supplementary Note 7). The fifth column shows the lowest-order basis functions for the corresponding irreducible representation.

symmetry. In the 2×2 unit cell, the mean-field Hamiltonians for CDW and SC orders are written as,

$$\delta H_{\text{MF}}^{\text{CDW}}[\boldsymbol{\Phi}] = \sum_{\boldsymbol{k},\sigma}\left[\Phi_{\text{AB}}\left(\tilde{A}_{\boldsymbol{k}\sigma}^\dagger O_1(\boldsymbol{k})\tilde{B}_{\boldsymbol{k}\sigma}\right) + \Phi_{\text{AC}}\left(\tilde{A}_{\boldsymbol{k}\sigma}^\dagger O_2(\boldsymbol{k})\tilde{C}_{\boldsymbol{k}\sigma}\right) \right.$$
$$\left. + \Phi_{\text{BC}}\left(\tilde{B}_{\boldsymbol{k}\sigma}^\dagger O_3(\boldsymbol{k})\tilde{C}_{\boldsymbol{k}\sigma}\right) + \text{C.C.}\right], \quad (5)$$

and

$$\delta H_{\text{MF}}[\Delta] = \Delta\left[\sum_{\boldsymbol{k}}\tilde{\Psi}_{\boldsymbol{k}\uparrow}^\dagger \hat{\Gamma}(\boldsymbol{k})\tilde{\Psi}_{\boldsymbol{k}\downarrow}^* + \text{C.C.}\right], \quad (6)$$

with the twelve-component spinor, $\tilde{\Psi}_{\boldsymbol{k}}^T = (\tilde{A}_{\boldsymbol{k}}^T, \tilde{B}_{\boldsymbol{k}}^T, \tilde{C}_{\boldsymbol{k}}^T)$ and $\tilde{\alpha}_{\boldsymbol{k}}^T = (\alpha_{1,\boldsymbol{k}}, \alpha_{2,\boldsymbol{k}}, \alpha_{3,\boldsymbol{k}}, \alpha_{4,\boldsymbol{k}})$. The indices $i \in \{1, 2, 3, 4\}$ and $\sigma \in \{\uparrow, \downarrow\}$ denote the orbital sites and spins, respectively. The specific forms of 4×4 bond matrices $O_a(\boldsymbol{k})$ and 12×12 pairing gap functions $\hat{\Gamma}(\boldsymbol{k})$ are tabulated in Table 1, 2. In our mean-field ansatz, the tested CDW order parameters are restricted to the 2×2 CDW orders carrying $l = 1$ angular momentum where the hopping strength modulates with the Q-vectors connecting $M_1, M_2, M_3$ points in the real space,

$$\Phi_{\text{AB}} = \sum_{\boldsymbol{R}}\cos(\boldsymbol{M}_3 \cdot \boldsymbol{R})\langle A_{\boldsymbol{R}}^\dagger B_{\boldsymbol{R}} - A_{\boldsymbol{R}}^\dagger B_{\boldsymbol{R}-2\boldsymbol{r}_1}\rangle \equiv \Phi\sigma_{\text{AB}}, \quad (7)$$

$$\Phi_{\text{AC}} = \sum_{\boldsymbol{R}}\cos(\boldsymbol{M}_2 \cdot \boldsymbol{R})\langle A_{\boldsymbol{R}}^\dagger C_{\boldsymbol{R}} - A_{\boldsymbol{R}}^\dagger C_{\boldsymbol{R}-2\boldsymbol{r}_2}\rangle \equiv \Phi\sigma_{\text{AC}}, \quad (8)$$

$$\Phi_{\text{BC}} = \sum_{\boldsymbol{R}}\cos(\boldsymbol{M}_1 \cdot \boldsymbol{R})\langle B_{\boldsymbol{R}}^\dagger C_{\boldsymbol{R}} - B_{\boldsymbol{R}}^\dagger C_{\boldsymbol{R}+2\boldsymbol{r}_3}\rangle \equiv \Phi\sigma_{\text{BC}}, \quad (9)$$

where the overall amplitude $\Phi$ quantifies the modulation strength and the relative phase of $(\sigma_{\text{AB}}, \sigma_{\text{AC}}, \sigma_{\text{BC}}) \in \{\pm 1, \pm i\}$ determines the pattern of CDW. Here, we fix the order parameter $\phi \in (\Phi, \Delta)$ is set as a real value. After constructing the ground state of the mean-field Hamiltonian $|G, \phi\rangle$ and evaluating the ground state energy $E[\phi] \equiv \langle G; \phi|H_0 + H_{\text{int}}|G; \phi\rangle$, we obtain the phase diagrams presented in Fig. 4 (see Supplementary Notes 8 and 9 for details). 161 grids of the ground energy with $\phi_i \in [-0.08, 0.08]$ are utilized and their numerical integration at given $\phi_i$ are carried out with 80×80 $\boldsymbol{k}$-points.

## First-principles calculations

We perform density-functional theory (DFT) calculations using the Vienna *abinitio* simulation package (VASP)[57,58] with the projector-augmented wave method[59]. For the exchange-correlation energy, the generalized-gradient approximation functional of Perdew-Burke-Ernzerhof[60] is employed. The van der Waals correction is included within the zero damping DFT-D3 method of Grimme[61]. The kinetic energy cutoff for the plane wave basis is 300 eV. The force criteria for optimizing the structures is set to 0.01 eV/Å. The monolayer $A$V$_3$Sb$_5$ is simulated using a periodic supercell with a vacuum spacing of ~ 20 Å. The $\boldsymbol{k}$-space integration is done with 10×17 $\boldsymbol{k}$-points for the $\sqrt{3}\times1$ structure. For the DOS calculation, we use 46×69 $\boldsymbol{k}$-points. The exfoliation energy is calculated using the Jung-Park-Ihm method[54]. We employ the finite difference method to obtain phonon dispersions. For the pristine monolayer (Fig. 3e), we use 4×4 supercell with 9×9 $\boldsymbol{k}$-points with the Phonopy software[62]. For the ISD-1 phase (Fig. 6b), we use 8×8 supercell with 6×6 $\boldsymbol{k}$-points using the nondiagonal supercell method[63]. We evaluate the (U, V) values in monolayer by using the first-principles constrained random phase approximation (cRPA) method[49] with the weighting approach[64], as implemented in VASP.

## Berry curvature and anomalous Hall conductivity

The berry curvature $\Omega_n(\boldsymbol{k})$ associated with the $n$-th energy band of the TB Hamiltonian $H(\boldsymbol{k})$ is given by

$$\Omega_n(\boldsymbol{k}) = -2\sum_{m\neq n}\frac{\text{Im}\langle u_{n\boldsymbol{k}}|\partial_x H(\boldsymbol{k})|u_{m\boldsymbol{k}}\rangle\langle u_{m\boldsymbol{k}}|\partial_y H(\boldsymbol{k})|u_{n\boldsymbol{k}}\rangle}{[E_n(\boldsymbol{k}) - E_m(\boldsymbol{k})]^2}. \quad (10)$$

Here, $E_n(\boldsymbol{k})$ and $|u_{n\boldsymbol{k}}\rangle$ are $n$th eigenvalue and eigenstate of $H(\boldsymbol{k})$ and the derivative in the momentum space $\partial_i \equiv \frac{\partial}{\partial k_i}$ is adopted. The anomalous Hall conductivity is calculated as a function of energy $E$ by integrating the $n$-band Berry curvatures for $E_n < E$ over the BZ

$$\sigma_{xy}(E) = \frac{1}{2\pi}\frac{e^2}{h}\int d^2k \sum_{E_n < E}\Omega_n(\boldsymbol{k}). \quad (11)$$

The numerical integration is performed by using 500×500 $\boldsymbol{k}$ points.

## Data availability

The data that support the findings of this study are available within the paper and Supplementary Information. Additional relevant data are available from the corresponding authors upon request.

## Code availability

The code that supports the findings of this study is available from the corresponding authors upon request.

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

## Acknowledgements

This work was supported by the Korean National Research Foundation (NRF) Basic Research Laboratory (NRF-2020R1A4A3079707). Y.K. acknowledges the support from the NRF Grant numbers (NRF-2021R1A2C1013871, NRF-2021M3H3A1038085). E.-G.M. acknowledges the support from the NRF funded by the Ministry of Science and ICT (No. 2019M3E4A1080411, No. 2021R1A2C4001847, No. 2022M3H4A1A04074153), National Measurement Standard Services and Technical Services for SME funded by Korea Research Institute of Standards and Science (KRISS -2022 - GP2022-0014). The computational resource was provided by the Korea Institute of Science and Technology Information (KISTI) (KSC-2020-CRE-0108) and the Cambridge Tier-2 system operated by the University of Cambridge Research Computing Service and funded by EPSRC [EP/P020259/1].

## Author contributions

E.G.M. and Y.K. designed and organized the research. S.W.K. and H.O. performed all calculations. E.G.M. and Y.K. supervised the research. All authors discussed the results and contributed to writing the manuscript.

## Competing interests

The authors declare no competing interests.
