## [Peer Review File · Nature Communications]

Monolayer Kagome Metals AV₃Sb₅REVIEWER COMMENTS

Reviewer #1 (Remarks to the Author):

The present manuscript has studied monolayer Kagome metals AV₃Sb₅ (A=K, Rb, Cs) using first-principles DFT and mean-field theory. The manuscript claims that monolayer AV₃Sb₅ can be formed with different symmetry than its parent bulk compound as a theoretical prediction. Also, it may show s- or d-wave superconductivity. It is well-known that for the past two years the AV₃Sb₅ compounds have attracted lots of attention with its rich physics of charge density wave and superconductivity. There are already many literatures on the bulk AV₃Sb₅ and less on monolayer. Being a layered material, it is not surprising that exfoliation of monolayer would not be so difficult in practice. I have many questions regarding current stage of this manuscript before the consideration of publication in nature communication.

(a) Whether the monolayer AV₃Sb₅ is thermodynamically stable and dynamically stable that should be decided by the Phonon bands plot and formation energy calculation. I cannot find the plots in present manuscript for all the AV₃Sb₅.

(b) The plot U vs V does show multiples phases in the monolayer system. But can you justify how those phases can be observed in experiment? How an experimentalist can tune U and V to get into those phases?

(c) CDW is already confirmed to exist in the bulk counterpart of the AV₃Sb₅. Although monolayer of the material has lower symmetry the existence of unconventional CDW phase is similar to its bulk counterpart. So, is there are justification behind this to mention in the present manuscript?

(d) In the bulk AV₃Sb₅ there is observation of CDW without acoustic phonon anomaly as reported in Phys. Rev. X 11, 031050 (2021). Have you found such situation also in monolayer cases?

I must accept the manuscript has given an excellent theoretical study to report exotic phases in monolayer AV₃Sb₅. But, the present form of manuscript need some clarifications in the description parts of the analysis before consideration for the publication in nature journal.

Reviewer #2 (Remarks to the Author):

In the manuscript "Monolayer Kagome Metals AV₃Sb₅", the authors present a monolayer realisation of the three-dimensional compound AV₃Sb₅. This novel kagome material has attracted considerable interest recently, showing an unconventional charge order at high and superconductivity at low temperatures. Due to the quasi-2D structure of AV₃Sb₅, a natural continuation is the investigation of systems of few layers. In the present manuscript, the authors investigate by the use of density functional theory how a stable monolayer configuration can be realised. Following this, a minimal tight-binding model is derived, allowing to investigate various electronic instabilities leading to a rich phase diagram.

The manuscript is well written and structured, however, while the numerical results are relevant and interesting, they are quite straightforwardly to be obtained. In my opinion, the manuscript is currently lacking concise interpretations of the numerical results. As bulk AV₃Sb₅ allows for intuitive explanations for most of the observed phenomena, the same should be possible for the monolayer. Especially the discussion of why and how these understandings change would significantly improve the quality of the manuscript and define the impact of the present work. Specifically, the following points should be addressed:

1 - The authors introduce type-II van Hove singularities appearing at the P points, however it is not really explained why they constitute a different type as the one at Gamma, beyond the sublattice constitution. Could the authors elaborate on this distinction?

2 - The authors derive a minimal model to describe the bands around the Fermi level, however the model is not motivated beyond the agreement with some DFT irreducible representations. In order to properly justify the model, the relevant orbitals contributing to bands around the Fermi level should be

discussed. Moreover, the band structure shows additional van Hove singularities close to the Fermi level not captured by the minimal model. Why can one exclude these bands?

3 - For the different charge orders, the form of the order parameter in real and/or momentum space is missing, which would allow for a classification in terms of angular momentum. One of the unique features of bulk AV₃Sb₅ is that the charge order potentially realises a non-zero angular momentum state, dubbed unconventional. Can one expect a similar behaviour in the monolayer?

4 - The phase diagram of monolayer AV₃Sb₅ is obtained by calculating the groundstate energy of a range of orders for the given parameters, however additional details are missing. The order parameter should at least be calculated self-consistently, it would be great if the authors could expand the discussion currently used in the main part and supplementary material.

5 - The emergence of a correlated electronic phase in bulk AV₃Sb₅ is attributed to the sublattice interference mechanism of the van Hove singularities, causing local interactions to be suppressed. This allows for the formation of long range interactions, promoting unconventional orders. In the monolayer case, this mechanism seems to be suppressed or even absent, leading to the question what mechanism can drive correlations now. In order to justify electronic instead of phonon driven phenomena, the authors should discuss this point.

Reviewer #3 (Remarks to the Author):

The authors propose a monolayer kagome material AV₃Sb₅. Enforcing charge neutrality, the authors performed density functional theory (DFT) calculations and found that the proposed monolayer has a different D2h structure than the bulk kagome metal AV₃Sb₅. Although the V atoms form a kagome net, the alkali atoms form two possible rectangular $\sqrt{3}\times 1$ structures. The authors suggest that the monolayer is stable and can be synthesized. In practice, this may not be that simple and may depend on the substrate conditions, etc. In particular, the monolayer charge neutrality condition may not hold as interface charge transfer can stabilize the more energetically favorable hexagonal AV₃Sb₅ monolayer.

Leaving this issue aside, the manuscript is evaluated based on whether the proposed D2h monolayer offers timely new physics of its own and/or new possibilities for exploring the new physics of the D6h bulk kagome metals. Unfortunately, as it stands, the manuscript and findings fall significantly short on both accounts and do not reach the level to meet the standard for publication in Nature Communications.

I list some concerns and comments below.

(1) The authors presented the DFT calculations for the crystal structure and the electronic band structure in the main text, summarized in Figs. 1 and 2. They find a reconstruction of the van Hove points in the rectangular zone for the D2h monolayer. Compared to the D6h structure, the van Hove singularity (VHS) at M₂ and M₃ points splits into four at the P-points away from the high-symmetry locations. Did the authors calculate the DFT phonon spectrum to determine the structural stability? Are there phonon softening, as observed in the systematic DFT studies of the bulk AV₃Sb₅ by Tan et al (Ref. [17])?

(2) For bulk AV₃Sb₅ with D6h symmetry, phonon softening of the breathing modes occurs at the M (and L) points (Ref. [17]), which is the basis for considering the 2x2 SD and ISD distortions. In the case of the D2h monolayer, what is the reason for considering such distortions? Why would one expect phonon softening to still happen at the 2x2 wave vectors? Moreover, since the VHS has been shifted to the four P-points, there is very little reason to expect a hexagonal 2x2 distortion. Is the distortion driven solely by phonons? Does the VHS play any part? What about the effects of electron-phonon coupling?

Although the supplemental contains some results of the phonon spectrum under the ISD distortion, a systematic DFT study of the monolayer D2h AV₃Sb₅ is necessary and even more desirable than jumping directly to the much discussed hexagonal states detected and studied for the different-in-symmetry bulk material.

(3) The authors instead fitted the DFT bands by a one-orbital tight-binding (TB) model with sublattice potentials and hopping integrals that break the D6h to D2h as a result of the $\sqrt{3}\times 1$ lattice potential. The TB model reasonably describes a set of low-energy bands and the VHS points labeled by P. A mean-field type of analysis is then conducted using a model interacting Hamiltonian given in Eq.(1). The interaction parameters U and V are taken to vary between attractions (<0) and repulsions (>0). I don't understand how the authors justify the consideration of such interactions. Some discussions must be in order, because it is confusing as the model stands.

(4) For $U, V > 0$, one can perhaps think of Eq.(1) as an extended Hubbard model. The possible electronic phases of this model on the pristine kagome lattice have been studied using well-controlled weak coupling theories at van Hove filling by Wang et al PRB 87, 115135 (2013). The authors should cite this work. Has a similar analysis been done for the TB model under Eq.(1)? Do the authors have any concrete predictions for the possible electronic phases of the D2h monolayer using the TB model?

(5) The new physics here seems to be the new type VHS points at P. Have the authors calculated the static susceptibility at different momenta to determine the possible instability in the particle-hole channel? The pristine kagome lattice at the van Hove filling has a divergent susceptibility at momenta connecting the VHS at the M points, which is behind the electronic instability of the 2×2 3Q CDW. The same cannot be said here without a careful study since the P points are away from the M points. How does the symmetry favor a 2×2 CDW such as the SD or the ISD?

(6) Since the VHS at the P points are away from any high symmetry points, have the authors considered the possibility of incommensurate CDW?

(7) It seems that the authors simply restricted the mean-field to the 2×2 CDW states studied extensively for the D6h kagome bulk material. This limits the ability to reveal the new physics of the monolayer with a different symmetry, and especially the intrinsic symmetry breaking effects due to the $\sqrt{3}\times 1$ structure. Moreover, how is such an approach justified? Doesn't the DFT already take into account at least part of the Coulomb interaction?

(8) The authors found TRSB flux phases when the chemical potential is moved close to the type-I VHS, which derives from the two VHS points at M₁ least affected by the $\sqrt{3}\times 1$ folding. This is interesting. The authors calculated the Chern numbers associated with each band in Fig. 4. This is the only figure that offered some details of the band structure in the symmetry breaking phases of the monolayer. They also calculated the anomalous Hall conductivity. There is a similar study on the pristine kagome lattice by Zhou et al arXiv:2110.06266. The TRSB current patterns look very similar. Are there magnetic fluxes threading the rest of the triangles? The authors should discuss the physics in more detail and compare to the case studied for the pristine kagome lattice. Fig. 4a indicates different Fermi pockets at M1 and M2, which are different from Zhou et al and presumably come from D6h symmetry breaking. The authors may be able to provide some experimental signatures for distinguishing the bulk and monolayer materials.

Summary of changes

Main text :

- We have newly added the section “instability of the monolayer”, highlighting the major innovation of the revised manuscript (Pages 6-8, vi-viii).
 - This section includes a new figure (Fig.3) of electronic susceptibility and phonon bands of the pristine monolayer.
 - This section elaborates on the instability to make a connection between the VHSs of the pristine monolayer and the order parameters introduced in the following section.
 - This section justifies the consideration of 2×2 CDW orders at non-zero temperatures.
 - This section justifies the consideration of incommensurate CDW orders at zero temperature.
 - This section justifies the introduction of long-ranged interaction and unconventional order parameters in the monolayer.
- We have cited the new references [Wang et al. PRB 87, 115135 (2013)] as Ref. [44] and added more discussion about the limitations of our analysis in the revised manuscript (page 10, x).
- We have compared our flux phases and their properties with those studied in the bulk [Zhou and Wang, arXiv:2110.06266] to show the essential physics of the monolayer (page 12, xii).
- We have added a paragraph to demonstrate the interplay physics between incommensurate CDW (IC-CDW) and type-II VHS in the “Discussion and Conclusion” section (page 13, xiii).
- We have discussed the prospects to elucidate the origin of CDWs in the “Discussion and Conclusion” section (page 13, xiii).
- We have added phonon bands of CDWs and their cohesive energy profiles in Fig. 6 to show the dynamical and thermodynamical stability and expanded discussion (Page 14, xiv).
- We have revised Table I and expanded the details about the order parameters to show a non-zero angular momentum state and unconventional CDWs in the “Methods” section (Page 16, xvi).

Supplementary information :

- We have added Fig. S6 in Supplementary Note 3.2. to elaborate on the difference between the type-I and -II VHS points.
- We have added Fig. S7 in Supplementary Note 3.2. and expanded relevant discussion to show non-universal features for the number of type-II VHS points.
- We have elucidated the role of sublattice interference by calculating sublattice-resolved bare electronic susceptibility in Supplementary Note 4.
- We have explained the symmetry-adapted 2×2 $3Q$ CDW order parameters in the monolayer compared to the bulk in Fig. S9. of Supplementary Note 5.1.
- We have included the band structures of various symmetry-breaking CDW phases in Fig. S10 of Supplementary Note 5.2.
- We have added Fig. S11 and Table S2 in Supplementary Note 5.3 to provide an in-depth comparison between our flux phases and their properties with those studied in the bulk [Zhou and Wang arXiv:2110.06266].
- We have added orbital-projected bands in Fig. S13 of Supplementary Note 6.3 to show relevant orbitals contributing to bands around the Fermi level and thus justification of our minimal model.
- We have shown the equivalence between the self-consistent equation and our variational method in Fig. S15. of Supplementary Note 7.4.
- We have justified the consideration of interactions and their origin used in mean-field phase diagrams in Supplementary Note 7.5.
- We have performed ab-initio constrained random phase approximation calculations to show U and V tunability under strains in Supplementary Note 9.
- We have added a discussion of incommensurate CDW in Supplementary Note 10. We have generalized the mean-field ansatz, including all possible IC-CDW constructed by connecting the shifted type-II VHSs, and showed the advanced phase diagram, Fig. S19. based on the 20×32 supercell calculation.

Answers to the 1st Referee's comments

Referee (R) : The present manuscript has studied monolayer Kagome metals AV_3Sb_5 ($A=K, Rb, Cs$) using first-principles DFT and mean-field theory. The manuscript claims that monolayer AV_3Sb_5 can be formed with different symmetry than its parent bulk compound as a theoretical prediction. Also, it may show s- or d-wave superconductivity. It is well-known that for the past two years the AV_3Sb_5 compounds have attracted lots of attention with its rich physics of charge density wave and superconductivity. There are already many literatures on the bulk AV_3Sb_5 and less on monolayer. Being a layered material, it is not surprising that exfoliation of monolayer would not be so difficult in practice. I have many questions regarding current stage of this manuscript before the consideration of publication in nature communication.

Authors (A) : We thank the Referee for the amazing summary of our manuscript. We also thank the Referee for raising important questions that greatly help improve our manuscript.

R: (a) Whether the monolayer AV_3Sb_5 is thermodynamically stable and dynamically stable that should be decided by the Phonon bands plot and formation energy calculation. I cannot find the plots in present manuscript for all the AV_3Sb_5 .

A: We thank the Referee for suggesting the plots that exhibit the thermodynamic and dynamic stability of the monolayer. Following the suggestions, we newly added the phonon bands, cohesive energy calculations, and energy plots in the revised manuscript as Fig. 6, showing the dynamical and thermodynamical stability, respectively.

In detail, the phonon bands of the pristine and ISD-1 CDW phases are shown, respectively, which are reproduced in Figs. R1**a** and **b**. The negative branches near Γ and M of the pristine phase in Fig. R1**a** disappear in the ISD-1 CDW phase as demonstrated in Fig. R1**b**. Therefore, the two plots clearly show the dynamical stability of the CDW phase. We note that the negligible negative branch near Γ in Fig. R1**b** occurs due to the computational conditions employed. These tiny negative frequencies can be eliminated by sampling more k points with an extended unit cell, required in diverse (quasi) two-dimensional systems [R1–R4].

In addition, the revised manuscript includes a new cohesive energy profile in atomic configuration space in Fig. 6**a** (Fig. R1**c**). We evaluated the cohesive energy per atom of the monolayer E_{coh} , which also captures the thermodynamic stability, to avoid the complexity of the reference

FIG. R1. **Phonon bands and cohesive energy of the AV_3Sb_5 monolayer.** **a** Phonon bands of the pristine KV_3Sb_5 monolayer. **b** Phonon bands of the ISD-1 KV_3Sb_5 monolayer. In the inset, the solid and dashed lines represent the BZs for the 2×2 and $\sqrt{3} \times 1$ structures, respectively. **c** Cohesive energy difference (per atom) as a function of the CDW distortion of the KV_3Sb_5 monolayer. The cohesive energy difference is defined as $\Delta E_{\text{coh}}(d) = E_{\text{coh}}(d=0) - E_{\text{coh}}(d)$, where the positive (negative) value of the distortion d is the maximum shift of the V atom in the ISD-1 (SD-2) phase with respect to the pristine structure.

states of constituent elements associated with the formation energy calculations by using the formula,

$$E_{\text{coh}} = [n_A E_A + n_V E_V + n_{\text{Sb}} E_{\text{Sb}} - E_{\text{tot}}] / (n_A + n_V + n_{\text{Sb}}). \quad (\text{R1})$$

Here, the number of atoms for each element n_A , n_V , and n_{Sb} is 1, 3, and 5, respectively, and E_A , E_V , E_{Sb} , and E_{tot} are the total energies of a single A atom, a single V atom, a single Sb atom, and monolayer, respectively. By definition, cohesive energy refers to the energy required to separate the crystals into isolated atoms. Thus, the higher the cohesive energy, the more thermodynamically stable the crystal. The cohesive energy of the monolayer is calculated as ~ 3.8 eV/atom for all three alkali atoms ($A = K, \text{Sb}, \text{and Cs}$). This value is comparable to the cohesive energy of the bulk system (~ 3.9 eV/atom). Therefore, the calculated cohesive energy of the monolayer supports thermodynamical stability. Furthermore, in Fig. 6a (Fig. R1c), we plotted the cohesive energy difference of the ISD-1 and SD-2 phases $\Delta E_{\text{coh}}(d) = E_{\text{coh}}(d=0) - E_{\text{coh}}(d)$ as a function of the distortion of V atoms d . It clearly shows that the CDW phases form a local minimum in the configuration space with an energy barrier of around four meV/atom. We believe that these plots demonstrate the thermodynamic stability of the monolayer.

R: (b) The plot U vs V does show multiples phases in the monolayer system. But can you justify how those phases can be observed in experiment? How an experimentalist can tune U and V to get into those phases?

A: We are very grateful for raising the insightful and substantive question. The tunability of U and V is one of our main motivations for the monolayer kagome metals. It is well established that various methods such as employing different substrates, gate dielectrics, or gate configurations may easily control effective Coulomb interactions [R5–R7]. We find that mechanical strains properly tune the U and V values. By using the first-principles constrained random phase approximation (cRPA) method [R8], we evaluated the (U, V) values for various strains shown in Fig. R2a. Our calculations reveal that the U value changes from 0.55 eV to 0.67 eV from 0 to 4 percents uniform tensile strains. Similarly, the V value varies from 0.07 eV to 0.11 eV under the same modulation of tensile strain.

Encouragingly, within these changes of U and V values, diverse phases are accessible. For example, for the chemical potential $\mu = 10$ meV, two different superconducting phases with d -wave (B_{1g}) and p -wave (B_{2u}) symmetries are accessible as shown in Fig. R2b. In addition, for the chemical potentials $\mu = 20$ and 30 meV, the SD-1 and TRSB-2 CDW and superconducting phases are observable within the ranges of the U and V values, as shown in Figs. R2c-d. Therefore, we believe that diverse phases can be accessed in experiments via strain engineering.

We added the following words to address the Referee questions in the revised manuscript (Page 12, xii):

“... For example, by applying a uniform biaxial strain in a range of $\pm 2\%$ variations, μ can be tuned from 50 meV to -100 meV (Fig. 5c). We also find that U and V values are functions of applied strains by performing the *ab initio* calculations with constrained random-phase approximation [49]. This indicates that different phases in the mean-field phase diagrams are accessible using strains (See Supplementary Note 9 for the detailed discussion and computational methods).

Moreover, ...”

FIG. R2. **a** U and V values as a function of uniform tensile strain. **b-d** The phase diagrams in U and V space for various chemical potentials. **b** $\mu = 10$ meV **c** $\mu = 20$ meV, and **d** $\mu = 30$ meV. The red boxed regions show the accessible phases under strain engineering.

R: (c) CDW is already confirmed to exist in the bulk counterpart of the AV_3Sb_5 . Although monolayer of the material has lower symmetry the existence of unconventional CDW phase is similar to its bulk counterpart. So, is there are justification behind this to mention in the present manuscript?

A: We thank the Referee for asking this crucial question. Yes, strong justifications exist. Our affirmative answer to this question highlights the major innovation of the revised manuscript. To address this question, we have newly performed 1) electronic susceptibility, 2) phonon energy bands, 3) sublattice-resolved susceptibility, and 4) energies of the incommensurate CDW orders. Based on these extensive calculations, we have added a new section in the revised manuscript to justify the introduction of the unconventional CDW phase similar to the bulk counterpart. Despite the distinct symmetry class, in which the monolayer lies, all these calculations point to the CDW phase similar to the bulk in the monolayer.

To summarize the results of the calculations, we first found that the electronic susceptibility generates a significant peak at $q = M$ for a wide range of electron filling. Thus, 2×2 CDW orders are necessitated to consider. Second, the phonon bands give rise to phonon softening at $q = M$, which indicates the 2×2 distortion of V atoms that generates the 2×2 CDWs. Third, an unconventional CDW phase is also expected from the significant interference between the sublattices, observed from our sublattice-resolved susceptibility calculations. Fourth, we found that the commensurate 2×2 CDWs are energetically favored over incommensurate CDWs in a finite region of the U and V parameter space. Finally, the celebrated Mermin-Wagner theorem dictates the dominance of the 2×2 CDWs at finite temperatures as it warrants the absence of continuous symmetry-broken phases in lower dimensions. Due to this theorem, the 2×2 CDWs, which break discrete translational symmetry, are more likely to be observed in experiments in the 2D monolayer than the incommensurate CDWs, which break continuous symmetry.

Answering the Referee's question, we wrote a new section 'Instability of the monolayer' and produced a direct answer to the question "*Can the unconventional CDW phase survive in the monolayer AV_3Sb_5 ?*". The unconventional CDW phase is one crucial issue in kagome metal physics. Our calculations clearly show the stability of the unconventional CDW phase in the monolayer, which may be similar to its bulk counterpart, as the Referee mentioned. Yet, we believe the stability issue is not trivial in the monolayer because the monolayer AV_3Sb_5 contains many new features, such as the different types of the VHSs, and we believe the inclusion of the discussion about the unconventional CDWs would be helpful to the community of the kagome metal physics. We sincerely thank the Referee for drawing this discussion.

(d) In the bulk AV_3Sb_5 there is observation of CDW without acoustic phonon anomaly as reported in Phys. Rev. X 11, 031050 (2021). Have you found such situation also in monolayer cases?

A: We thank the Referee for providing insight regarding the origin of CDW. Unfortunately, to our best knowledge, the current theoretical study did not generate any supporting or opposing indications of such a situation where CDW occurs without acoustic phonon anomaly. We believe that the observation of CDW without acoustic phonon anomaly is crucial to resolving the CDW mechanism of the monolayer as excellently discussed in the bulk case [R9, R10]. In the revised manuscript, we pointed to the literature and highlighted the important insight provided by the Referee in our revised manuscript for future experimental and theoretical studies by adding the following sentences in the “Discussion and Conclusion” section (Page 14, xiv):

“... In this regard, further studies in the monolayer should be complementary in pinning down the origin of the CDW orders. The observation of CDW without acoustic phonon anomaly and evolution of CDW amplitude modes may be important to resolve the CDW mechanism of the monolayer as discussed in the bulk case [52,53] ...”

I must accept the manuscript has given an excellent theoretical study to report exotic phases in monolayer AV_3Sb_5 . But, the present form of manuscript need some clarifications in the description parts of the analysis before consideration for the publication in nature journal.

A: We are very grateful for the Referee’s positive assessment of our study. We believe that the revised manuscript can be positively considered for publication in Nature Communications.

Answers to the 2nd Referee's comments

In the manuscript “Monolayer Kagome Metals AV_3Sb_5 ”, the authors present a monolayer realisation of the three-dimensional compound AV_3Sb_5 . This novel kagome material has attracted considerable interest recently, showing an unconventional charge order at high and superconductivity at low temperatures. Due to the quasi-2D structure of AV_3Sb_5 , a natural continuation is the investigation of systems of few layers. In the present manuscript, the authors investigate by the use of density functional theory how a stable monolayer configuration can be realised. Following this, a minimal tight-binding model is derived, allowing to investigate various electronic instabilities leading to a rich phase diagram.

A: We thank the Referee for the nice summary of our manuscript.

The manuscript is well written and structured, however, while the numerical results are relevant and interesting, they are quite straightforwardly to be obtained. In my opinion, the manuscript is currently lacking concise interpretations of the numerical results. As bulk AV_3Sb_5 allows for intuitive explanations for most of the observed phenomena, the same should be possible for the monolayer. Especially the discussion of why and how these understandings change would significantly improve the quality of the manuscript and define the impact of the present work. Specifically, the following points should be addressed:

A: We appreciate the comment and suggestion of the Referee. We are delighted to see the Referee's comment, “The manuscript is well written and structured”. The Referee raised thoughtful concerns about physical interpretations of our results. Preparing the revision, we have strengthened and expanded the discussion about the physical interpretations. We believe that our manuscript is significantly improved, and we hope that our detailed responses convince the Referee.

1 - The authors introduce type-II van Hove singularities appearing at the P points, however it is not really explained why they constitute a different type as the one at Gamma, beyond the sublattice constitution. Could the authors elaborate on this distinction?

A: This is a very important comment, for which we are grateful to the Referee. We stress that the type-II VHS is associated with the hybridization between two different M points at the Brillouin zone boundary (say, M_2, M_3) while the type-I VHS at Γ is just a shift of one M point (say, M_1), as illustrated in Fig.R3. The symmetry-lowering of the monolayer system hybridizes the two bands with different sublattice constitutions at the zone boundary, represented by different colors (red, green in Fig.R3) and produces the four type-II VHSs. Note that the number of the resultant VHS points (four in Fig.R3) is not universal and depends on the microscopic details of the system, such as a Fermi-surface nesting condition parameter, as detailed in Supplementary Note 3.2.

In the revised manuscript, to elaborate on the distinction, we have added the following sentences in the main text (Page 4, iv) with a detailed analysis in Supplementary Note 3.2.

“... First, the appearance of the type-II VHS off time-reversal invariant momenta is understood as a generic result attributed to the band hybridization allowed by the symmetry lowering. Since the two VHSs at M_2 and M_3 before the symmetry lowering are at the exact same energy level, they can be mixed together and contribute to generating the type-II VHSs (See Fig. 1c). On the other hand, the VHS at M_1 still remains a type-I VHS because no states are available to hybridize with. We find that the number of the generated type-II VHS points (in this case, four) is not universal and depends on the microscopic details of the system (see details in Supplementary Note 3.2).
...”

FIG. R3. Schematic illustration of the type-I and -II VHS points in momentum space rearranged by symmetry lowering. **a,b.** The distribution of the VHS point under the $(T_{1 \times 1}, D_{6h})$ symmetry. A solid (dashed) black line indicates the Brillouin zone of the $\sqrt{3} \times 1$ (1×1) unit cell. **c,d.** The symmetry-lowering into $(T_{\sqrt{3} \times 1}, D_{2h})$ leads to the hybridization of the VHS states at M . As a result, the two VHSs at M become mixed and split into four type-II VHS points, consisting of a mixed sublattice character. On the other hand, the VHS at Γ remains as a type-I VHS with a single sublattice character.

R: 2 - The authors derive a minimal model to describe the bands around the Fermi level, however the model is not motivated beyond the agreement with some DFT irreducible representations. In order to properly justify the model, the relevant orbitals contributing to bands around the Fermi level should be discussed.

A: As the Referee suggested, we augmented the justification of our minimal model in terms of the relevant orbitals contributing to bands around the Fermi level. The orbital-projected bands shown in Fig. R4 reveal that the VHS band described by our model mainly comprises three d -orbitals of V atoms: d_{xy} , $d_{3z^2-r^2}$, and $d_{x^2-y^2}$. Our group-theoretical analysis of the DFT Bloch state at M shows that the VHS state is in the A_g irreducible representation of the little group of M . This indicates that the three d orbitals of V are hybridized to form a A_g orbital, based on which our minimal model is constructed. Therefore, our minimal model successfully reproduced the DFT VHS band. We added the orbital projected bands in Supplementary Fig.S13.

FIG. R4. **Orbital-projected band structures of the AV_3Sb_5 monolayer.** d_{yz} , d_{xz} , d_{xy} , $d_{3z^2-r^2}$ and $d_{x^2-y^2}$ orbitals of V atoms are projected on the band structure where the magnitude of open circles is proportional to the projected weight.

R: Moreover, the band structure shows additional van Hove singularities close to the Fermi level not captured by the minimal model. Why can one exclude these bands?

A: We thank the Referee for asking the question. The excluded van Hove singularities that are relatively away from the Fermi level. Thus, they can be considered less important in describing the low-energy electronic property of the AV_3Sb_5 monolayer. In other words, with well-screened Coulomb interactions at low enough temperatures, the minimal model is good enough to capture the physics of the AV_3Sb_5 monolayer.

We would like to point out one nice property of our minimal model. Namely, unlike the bulk materials [R11], the A_g van Hove singularity nearest to the Fermi level always appears in the monolayer, irrespective of the type of alkali metal A (Fig. R4), which is automatically captured by our minimal model.

Motivated by the Referee's comment, we added the above discussion in the Supplementary Note 6.3 with the orbital-projected band structures.

R: 3 - For the different charge orders, the form of the order parameter in real and/or momentum space is missing, which would allow for a classification in terms of angular momentum. One of the unique features of bulk AV_3Sb_5 is that the charge order potentially realises a non-zero angular momentum state, dubbed unconventional. Can one expect a similar behaviour in the monolayer?

A: We thank the Referee for addressing this point, which allows us to improve our manuscript. In Tables I and II of the previous manuscript, we provided information about the form of the order parameters in momentum space. Preparing the response to the Referee, we further found that all CDW orders can be expressed by linear combinations of the three charge bond orders,

$$\begin{aligned}\Phi_{AB} &= \sum_{\mathbf{k} \in 1 \times 1 \text{B.Z.}} \left[-2is_1(\mathbf{k}) \langle a_{\mathbf{k}}^\dagger b_{\mathbf{k}+\mathbf{Q}_3} \rangle + \text{C.C.} \right], & \Phi_{AC} &= \sum_{\mathbf{k} \in 1 \times 1 \text{B.Z.}} \left[-2is_2(\mathbf{k}) \langle a_{\mathbf{k}}^\dagger c_{\mathbf{k}+\mathbf{Q}_2} \rangle + \text{C.C.} \right], \\ \Phi_{BC} &= \sum_{\mathbf{k} \in 1 \times 1 \text{B.Z.}} \left[-2s_3(\mathbf{k}) \langle b_{\mathbf{k}}^\dagger c_{\mathbf{k}+\mathbf{Q}_1} \rangle + \text{C.C.} \right].\end{aligned}\quad (\text{R2})$$

Note that all the three order parameters are unconventional, carrying an angular momentum, $l = 1$, indicated by the form factor ($s_i(\mathbf{k}) \equiv \sin \mathbf{k} \cdot \mathbf{r}_i$). Here, \mathbf{r}_i is a distance vector between the nearest neighbors of sublattice sites.

Since this is an important point of our analysis, we have expanded the discussion about the

order parameters in the “mean-field theory” part in the “Methods” section with the revised Table I in the main text.

R: 4 - The phase diagram of monolayer AV_3Sb_5 is obtained by calculating the ground state energy of a range of orders for the given parameters, however additional details are missing. The order parameter should at least be calculated self-consistently, it would be great if the authors could expand the discussion currently used in the main part and supplementary material.

A: We would like to emphasize that our variational method is essentially equivalent to solving a self-consistency equation at zero temperature. The self-consistency equation is obtained by differentiating free energy with respect to the mean-field order parameter to find a stationary point. On the other hand, the variational method directly determines the minimum of the free energy by varying the order parameter of the wave function. Therefore, the equivalence between the two methods is well established, unless the solution is located at local minimum points. To check the equivalence, we plot the magnitudes of the order parameter carried out through the two methods in Fig. R5, where the perfect matching between the two methods is demonstrated.

Following the Referee’s suggestion, we have added the above discussions with Fig. S15 in the new section in Supplementary Note 7.4 to clarify our mean-field calculations.

FIG. R5. Order parameter (SD-2) amplitude of phase as a function of U and the corresponding phase diagram at the chemical potential $\mu = -6\text{meV}$. **a.** The amplitude of the SD-2 CDW order parameter for $U < 0$ and $V = 0$. The same results are produced from the self-consistency equation (SCE) and the variational method (VM) at zero temperature, demonstrating the equivalence of the two methods. **b.** The mean-field phase diagram contains the tested SD-2 CDW order. The black arrow in the inset indicates the considered U and V values. The chemical potential is given by $\mu = -6\text{meV}$.

R: 5 - The emergence of a correlated electronic phase in bulk AV_3Sb_5 is attributed to the sublattice interference mechanism of the van Hove singularities, causing local interactions to be suppressed. This allows for the formation of long range interactions, promoting unconventional orders. In the monolayer case, this mechanism seems to be suppressed or even absent, leading to the question what mechanism can drive correlations now. In order to justify electronic instead of phonon driven phenomena, the authors should discuss this point.

A: We thank the Referee for the comment. In monolayer AV_3Sb_5 , the sublattice interference also survives and plays a key role in the formation of long-range interactions (V), promoting unconventional orders. One can demonstrate the interference in the monolayer system by calculating the sublattice resolved susceptibility, following the previous literature [R12]. The explicit form of the susceptibility is given by

$$\chi^{l_1, l_2}(\mathbf{q}) \equiv - \sum_{\mathbf{k}, \mu, \nu} a_{\nu}^{l_1}(\mathbf{k} + \mathbf{q}) a_{\mu}^{l_1}(\mathbf{k})^* a_{\nu}^{l_2}(\mathbf{k} + \mathbf{q})^* a_{\mu}^{l_2}(\mathbf{k}) \left[\frac{n_F(E_{\mu}(\mathbf{k})) - n_F(E_{\nu}(\mathbf{k} + \mathbf{q}))}{E_{\mu}(\mathbf{k}) - E_{\nu}(\mathbf{k} + \mathbf{q})} \right] \quad (\text{R3})$$

with sublattice index, (l_1, l_2) , and band index, (μ, ν) . Here, the Fermi distribution, $n_F(\epsilon)$, and the l_i -th component of the eigenvector of the μ band, $a_{\mu}^{l_i}(\mathbf{k})$, are introduced. The eigenvector of the largest eigenvalue of $\chi^{l_1, l_2}(\mathbf{q})$ around M point encode the sublattice interference effects. The two important cases of the sublattice weights of (A, B, C) sublattice are,

1. the cases with the perfect sublattice interference : $(0, 0.5, 0.5)$, $(0.5, 0, 0.5)$, $(0.5, 0.5, 0)$,
2. the cases without the sublattice interference : $(1, 0, 0)$, $(0, 1, 0)$, $(0, 0, 1)$,

as illustrated in Fig. R6 **a**.

In the monolayer AV_3Sb_5 , as the Referee correctly pointed out, the sublattice interference is affected but remains significant. We found that the largest eigenvalues of the susceptibility at the M point have the sublattice weights, $(0.482, 0.258, 0.258)$ and $(0.476, 0.262, 0.262)$, for the type-I, type-II VHS filling, respectively. These values are significantly different from the the cases without the sublattice interference (Fig. R6**b,c**).

We further have estimated the values of U and V in the monolayer and compared them with their bulk values in Table R1 by employing the first-principles constrained random phase approximation (cRPA) calculation [R8]. The intersite interaction parameter V of our minimal A_g tight-binding model for monolayer AV_3Sb_5 is $V = 0.07$, which increased by about %20 percent compared to that of the bulk, $V = 0.06$. Hence, we argue that V still plays a crucial role in offering new

physics in monolayer AV_3Sb_5 , as in the bulk system.

The origin of the mechanism of correlation effects is certainly an important problem, but even in the bulk AV_3Sb_5 , the origin has not been resolved to our best knowledge. Thus, our finding of the sublattice interference of the monolayer AV_3Sb_5 strongly calls for future theoretical and experimental works.

In the revised manuscript, we added the following sentence on Page 6 (vi):

“... Moreover, the sublattice character of $\chi(q)$ reveals that sublattice interference is significant in the monolayer (See Supplementary Note 4 for the calculations of sublattice-resolved susceptibility). This result warrants the consideration of longer-range interactions, which promotes unconventional orders as in the bulk cases [6, 20, 43, 44]. ...”. We also newly added the detailed analysis of sublattice-resolved susceptibility calculation results in Supplementary Note 4.

	Monolayer		Bulk	
Target orbitals	$U(F_0)$ (eV)	V (eV)	$U(F_0)$ (eV)	V (eV)
All (bare)	14.20	3.93	15.79	4.46
V- d Sb- p	5.87	1.96	5.34	1.48
V- d	0.77	0.11	0.80	0.08
V- $d3$ (A_g orbital)	0.58	0.07	0.60	0.06

TABLE R1. **On-site U and nearest-neighbor V Hubbard interaction parameters calculated by cRPA method [R8].** In cRPA method, we divide the full polarization function into two parts as $P = P_t + P_r$ in which P_t and P_r are calculated within the target and rest orbitals, respectively. The effective Coulomb interaction matrix \mathcal{U} is computed via $\mathcal{U} = [1 - \mathcal{U}_{\text{bare}} P_r]^{-1} \mathcal{U}_{\text{bare}}$ where $\mathcal{U}_{\text{bare}}$ is a bare Coulomb interaction. We then average the matrix \mathcal{U} to obtain U, V parameters. The screened Hubbard parameters U and V in the bulk are quite similar to those of the previous study [R13]. The effective U and V values depend on the target orbital models. Our minimal A_g orbital TB model includes three V- d target orbitals of $d_{3z^2-r^2}$, $d_{x^2-y^2}$, and d_{xy} in global coordinates (Fig. R4), dubbed V- $d3$ model. Interestingly, due to the large screening effect, the U and V values are strongly renormalized as $U = 0.58$ and $V = 0.07$ eV for our minimal A_g orbital TB model.

FIG. R6. **Sublattice-resolved bare susceptibility of AV_3Sb_5 systems.** **a.** The three different sublattices A, B, and C are colored in blue, red, and green, respectively. It is useful to utilize the eigenvector associated with the largest eigenvalues of $\chi^{l_1, l_2}(\mathbf{q})$ to quantify the sublattice interference. The two important cases with (denoted by a circle) and without (denoted by a rectangle) the perfect sublattice interference are illustrated by comparing with our system (denoted by \times). The sublattice character of the monolayer AV_3Sb_5 system shows the intermediate behaviors between two limiting cases. **b-c.** The eigenvalues of bare susceptibility matrix $\chi^{l_1, l_2}(\mathbf{q})$ and their sublattice characters are carried out based on our tight-binding Hamiltonian at $\beta = 1000$. (b) and (c) show the results for the Type-I, and Type-II VHS filling of the AV_3Sb_5 systems. Focusing on the sublattice character of largest eigenvalues at $\mathbf{q} = M$, we find that $(0.482, 0.258, 0.258)$, and $(0.476, 0.262, 0.262)$ for (b),(c), respectively.

Answers to the 3rd Referee's comments

R: The authors propose a monolayer kagome material AV_3Sb_5 . Enforcing charge neutrality, the authors performed density functional theory (DFT) calculations and found that the proposed monolayer has a different D2h structure than the bulk kagome metal AV_3Sb_5 . Although the V atoms form a kagome net, the alkali atoms form two possible rectangular $\sqrt{3}\times 1$ structures. The authors suggest that the monolayer is stable and can be synthesized. In practice, this may not be that simple and may depend on the substrate conditions, etc. In particular, the monolayer charge neutrality condition may not hold as interface charge transfer can stabilize the more energetically favorable hexagonal AV_3Sb_5 monolayer.

A: We thank the Referee for providing invaluable insight into kagome materials AV_3Sb_5 . We agree that the experimental realization of a AV_3Sb_5 monolayer might depend on certain practical difficulties, such as the neutrality condition, as the Referee mentioned.

Yet, we also believe that it is imperative to investigate the new possibility and significance of the monolayer systems since many advantages may exist in the monolayer systems as manifested in graphene and two-dimensional van der Waals materials. For example, an IC-CDW order parameter is intrinsically prohibited at non-zero temperature based on the celebrated Mermin-Wagner theorem, and its strong fluctuations are expected. This may host new physics in the monolayer systems. We further believe that the unrevealed difficulties, such as the neutrality conditions associated with the substrate effect, may make the monolayer physics even richer. Our works may contribute to the monolayer physics of kagome metals. To emphasize our viewpoints, we add the following sentences to share the Referee's concern and insight in the revised manuscript:

R: Leaving this issue aside, the manuscript is evaluated based on whether the proposed D2h monolayer offers timely new physics of its own and/or new possibilities for exploring the new physics of the D6h bulk kagome metals. Unfortunately, as it stands, the manuscript and findings fall significantly short on both accounts and do not reach the level to meet the standard for publication in Nature Communications. I list some concerns and comments below.

A: We thank the Referee for sharing the time and effort to carefully read our paper and provide constructive comments, which greatly help improve the quality of the paper. Below, we have considered and modified all the points mentioned by the Referee. Revising the manuscript, we

believe that our manuscript is significantly improved, warranting a publication in Nature communications. We hope that the Referee agrees with us.

R: (1) The authors presented the DFT calculations for the crystal structure and the electronic band structure in the main text, summarized in Figs. 1 and 2. They find a reconstruction of the van Hove points in the rectangular zone for the D2h monolayer. Compared to the D6h structure, the van Hove singularity (VHS) at M2 and M3 points splits into four at the P-points away from the high-symmetry locations. Did the authors calculate the DFT phonon spectrum to determine the structural stability? Are there phonon softening, as observed in the systematic DFT studies of the bulk AV_3Sb_5 by Tan et al. (Ref. [17])?

A: We thank the Referee for the critical comment. To address the Referee's concerns, we performed a systematic study of the DFT phonon spectra of the monolayer. As the Referee predicted, we found phonon softening in the pristine monolayer, as in the bulk case (Ref. [17]). The DFT phonon spectra of the pristine monolayer for all three alkali atoms (Fig. R7) consistently show the phonon softening at the Γ and M points. These phonon spectra of the pristine monolayer indicate structural instability.

FIG. R7. Phonon bands of the pristine phases for the AV_3Sb_5 monolayer family. Here, the smearing factor $\sigma = 0.01$ eV is used.

R: (2) For bulk AV_3Sb_5 with D6h symmetry, phonon softening of the breathing modes occurs at the M (and L) points (Ref. [17]), which is the basis for considering the 2x2 SD and ISD distortions. In the case of the D2h monolayer, what is the reason for considering such distortions?

A: We thank the Referee for asking in-depth questions and providing insights regarding the structural instability. Like the bulk system, the $\sqrt{3} \times 1$ monolayer produces phonon softening at Γ and M points associated with the distortion of V atoms. We believe that this result justifies the consideration of the 2×2 SD and ISD distortions in the monolayer.

R: Why would one expect phonon softening to still happen at the 2x2 wave vectors? Moreover, since the VHS has been shifted to the four P-points, there is very little reason to expect a hexagonal 2x2 distortion.

A: Indeed it appears that the rearrangement of the VHSs could hinder the phonon softening at the 2×2 wave vectors. However, our static susceptibility calculations show it depends on electron filling. In particular, significant contributions are found from the M points in a wide range of electron filling, supporting the expectation of phonon softening at the 2×2 wave vectors.

The static susceptibility $\chi(\mathbf{q})$ is calculated using the below Eq.

$$\chi(\mathbf{q}) \equiv - \sum_{\mathbf{k}, \mu, \nu} \left[\frac{n_F(E_\mu(\mathbf{k})) - n_F(E_\nu(\mathbf{k} + \mathbf{q}))}{E_\mu(\mathbf{k}) - E_\nu(\mathbf{k} + \mathbf{q})} \right], \quad (\text{R4})$$

where $n_F(\epsilon)$ is the Fermi distribution and (μ, ν) is band index. Figure R8a-d shows $\chi(\mathbf{q})$ calculated at various fillings. We find that the presence of both type-I and type-II VHSs leads to diversified instability sensitively depending on electron filling. For example, at the type-II VHS filling (Fig. R8a), a significant contribution arises from the q vectors that connect the van Hove saddle points. The peaks at q_1 and q_2 in $\chi(\mathbf{q})$ correspond to the nesting vectors mediating distinct P_i points. These incommensurate q vectors motivate the consideration of an incommensurate order parameter, such as IC-CDW phases (we detailed this in the reply to comment 6 below). By contrast, when the chemical potential increases, the peaks quickly merge and give rise to the significant enhancement of $\chi(\mathbf{q})$ arises at the M point (Figs. R8b-d). This significant peak at $q = M$, as well as at $q = \Gamma$ in the folded BZ, corresponds to the commensurate 2×2 CDW vectors. Notably, the large values of $\chi(\mathbf{q})$ at M survive for a wide range of chemical potentials. These results suggest a significant role of 2×2 CDW orders in stabilizing the monolayer.

FIG. R8. **Electronic susceptibility and phonon bands of the AV_3Sb_5 monolayer.** **a-c** Electronic susceptibility $\chi(\mathbf{q})$ is calculated in 2D (q_x, q_y) space for **a** the type-II VHS filling ($\mu = -6$ meV), **b** neutral filling ($\mu = 0$ meV), and **c** type-I VHS filling ($\mu = 9$ meV). $\chi(\mathbf{q})$. The right panels show a magnified view of the boxed areas of the left panel near the high-symmetry Γ and M points, respectively. \mathbf{q}_1 and \mathbf{q}_2 in **a** represent the nesting vectors that connect the type-II VHS points (black circles). **d** Line profiles of $\chi(\mathbf{q})$ along the high-symmetry lines of the BZ for various chemical potentials. The corresponding chemical potential (in meV) is shown in each profile. The momenta correspond to \mathbf{q}_1 and \mathbf{q}_2 vectors in **a** are indicated by arrows with vertical lines. **e** Phonon bands of the pristine KV_3Sb_5 monolayer as a function of electronic temperature. The electronic temperature effect is introduced by tuning the smearing factor σ (from 0.1 to 0.3 eV) of the Fermi-Dirac distribution function.

R: Is the distortion driven solely by phonons? Does the VHS play any part? What about the effects of electron-phonon coupling?

A: To address the Referee's questions, we additionally calculated the phonon spectra as a function of electronic temperature, smearing factor σ , as shown in Fig. R8e. Based on these calculations, we conclude that both phonons and the VHS play some part in the distortion. The softened phonon at M is lifted as we destroy the Fermi surface by increasing the electronic temperature. These calculations demonstrate that the electronic instability associated with the VHS plays a role in the formation of structural distortion. Since the DFT phonon spectra include the effects of electron-phonon coupling, our results are suggestive that either electronic or phononic contribu-

tions cannot be excluded in forming structural instability. In addition, the occurrence of the 2×2 instability in the phonon spectra is plausible given the electronic susceptibility calculations. Our DFT calculations are performed off the type-II VHS filling, in which large susceptibility values arise near the M point.

R: Although the supplemental contains some results of the phonon spectrum under the ISD distortion, a systematic DFT study of the monolayer $D2h AV_3Sb_5$ is necessary and even more desirable than jumping directly to the much discussed hexagonal states detected and studied for the different-in-symmetry bulk material.

A: We agree with the Referee that we were rough in introducing the 2×2 distortions. Following the Referee's suggestion, we added a new section "Instability of the monolayer" in the main text with a detailed analysis based on the static susceptibility and phonon calculations with a relevant discussion about the instability of the monolayer and justification for the consideration of 2×2 commensurate order parameters. We sincerely thank the Referee for this suggestion, due to which our study has become more complete and innovative.

R: (3) The authors instead fitted the DFT bands by a one-orbital tight-binding (TB) model with sublattice potentials and hopping integrals that break the $D6h$ to $D2h$ as a result of the $\sqrt{3}\times 1$ lattice potential. The TB model reasonably describes a set of low-energy bands and the VHS points labeled by P. A mean-field type of analysis is then conducted using a model interacting Hamiltonian given in Eq.(1). The interaction parameters U and V are taken to vary between attractions (< 0) and repulsions (> 0). I don't understand how the authors justify the consideration of such interactions. Some discussions must be in order because it is confusing as the model stands.

A: We thank the Referee for asking the question. The screening of the long-range Coulomb interaction in metals is one of the reasons why we consider U and V in our mean field analysis with the effective tight-binding model. As in the famous Hubbard model, it is well established that the U term is usually enough to capture the interaction effects. However, in AV_3Sb_5 , the intriguing interference effects exist, so it is important to consider the next nearest interaction V . In principle, one can consider even the next-next nearest (or even further) interaction term, but the screening effect significantly suppresses further interactions. Therefore, it is reasonable to start with the interacting Hamiltonian, Eq. (1) in the main text. We also note that similar approaches have been widely used to investigate properties of bulk AV_3Sb_5 [R14].

By adopting the well-established first-principles constrained random phase approximation (cRPA) calculation [R8], we have estimated the values of (U, V) . In Fig. R9, we have plotted the values of (U, V) at fixed chemical potentials varying strains. The cRPA results show that quantum phase transitions can be driven by tuning strains, as illustrated in Fig.R9. We also note that electron-phonon interaction may effectively contribute to (U, V) and renormalize them toward negative values. These phonon-mediated attractive channels are known as a central origin of conventional superconductivity and have been taken into account in various theoretical works [R7, R15]. Regarding this point, we have added a few words in Supplementary Note 7.5 to clarify the origin of the attractive channel (U or $V < 0$).

FIG. R9. **a** U and V values as a function of uniform tensile strain. **b-d** The phase diagrams in U and V space for various chemical potentials. **b** $\mu = 10$ meV **c** $\mu = 20$ meV, and **d** $\mu = 30$ meV. The red boxed regions show the accessible phases under strain engineering.

(4) For $U, V > 0$, one can perhaps think of Eq.(1) as an extended Hubbard model. The possible electronic phases of this model on the pristine kagome lattice have been studied using well-controlled weak coupling theories at van Hove filling by Wang et al PRB 87, 115135 (2013). The authors should cite this work. Has a similar analysis been done for the TB model under Eq.(1)? Do the authors have any concrete predictions for the possible electronic phases of the D2h monolayer using the TB model?

A: We thank the Referee for letting us know the reference, [Wang et al. PRB 87, 115135 (2013)], where the singular-mode functional renormalization group (SMFRG) method was used. In our analysis, we performed only the standard mean-field analysis with the tight-binding model and the density functional calculations. We believe it is a plausible choice because our main purpose is to determine available phases of the monolayer AV_3Sb_5 systems. We believe that our mean-field analysis is strong enough to meet our purpose. Furthermore, it has successfully established the global shapes of the phase diagrams. In other words, our analysis is asymptotically correct when order parameters become large enough as is generically expected from the standard mean-field theory.

Yet, we agree with the Referee that proper treatment of correlation effects necessitates a more sophisticated and high-fidelity analysis, especially for weak coupling analysis or phase boundaries. In future works, we hope to incorporate correlation effects precisely by using more sophisticated methods such as SMFRG. We have cited the mentioned reference [Wang et al. PRB 87, 115135 (2013)] as Ref. [44] and added more discussion about the limitations of our analysis in the revised manuscript (Page 10, x):

“...where the zero chemical potential is set to the neutral filling. **We emphasize that our mean-field analysis of AV_3Sb_5 monolayer is reliable at zero and very low temperatures since any monolayer systems suffer from significant thermal fluctuations. Thus, the phase diagrams of Fig. 4 need to be understood as the ones in the limit of lowering temperature down to zero, $T \rightarrow 0^+$. In what follows ...**”

(5) The new physics here seems to be the new type VHS points at P. Have the authors calculated the static susceptibility at different momenta to determine the possible instability in the particle-hole channel? The pristine kagome lattice at the van Hove filling has a divergent susceptibility at momenta connecting the VHS at the M points, which is behind the electronic instability of the 2x2 3Q CDW. The same cannot be said here without a careful study since the P points are away from the M points. How does the symmetry favor a 2x2 CDW such as the SD or the ISD?

A: We much appreciate the Referee's constructive and substantive comments. In the revised manuscript, we newly added the calculations of the static susceptibility to determine the possible instability, which provided us with a new opportunity to encounter novel physics, as detailed in the above reply to comment 2.

Here, we additionally argue that the lowered symmetries of the monolayer can favor the 2×2 3Q SD and ISD phases as they respect the lower symmetries. A close look at the SD and ISD distortion of the monolayer reveals that they adapt themselves into the lowered symmetries by losing C_3 symmetries and being split into the doublets, which we refer to as SD-1 and SD-2 and ISD-1 and ISD-2, respectively. As delineated in Fig. R10, the adjacent two vanadium atoms have different bonding lengths between the AB and BC sublattices. These patterns are unlike the D_{6h} bulk counterpart, in which the bonding lengths are the same between the adjacent V atoms. Since these types of distortion respect the $\sqrt{3} \times 1$ symmetries, it is likely that the monolayer symmetry favors them.

FIG. R10. **Bond lengths of the ISD-1 phase in the monolayer.** Sublattice indices A, B, and C (for hexagons) are depicted on the left panel. The calculated bond lengths between the sublattices are given in Table (in \AA). In contrast to the isotropic 3Q CDW order of the bulk counterpart, the 3Q CDW in the monolayer has anisotropic bond lengths, such that $d_{BC} < d_{AB} = d_{AC}$.

We added the above discussions in the Supplementary Note 5.1.

(6) Since the VHS at the P points are away from any high symmetry points, have the authors considered the possibility of incommensurate CDW?

A: This is a keen observation, and, as the Referee mentioned, it seems natural to consider the incommensurate CDW (IC-CDW) order parameters from the Type-II VHS points (P_i).

Yet, we excluded IC-CDW order parameters in the previous manuscript because our systems have a monolayer structure. As proven by Mermin and Wagner as well as Hohenberg and Coleman [R16–R18], the IC-CDW order parameters in the monolayer structure are prohibited at any non-zero temperature in stark contrast to IC-CDWs in layered or three-dimensional structures. Moreover, the IC-CDW is much more vulnerable to impurities than commensurate CDW orders, and infinitesimally small amounts of impurities destabilize the IC-CDW [R19]. Thus, the IC-CDW order parameters in monolayer systems can usually be ignored, and we decide to focus on the commensurate CDW order parameters in the main text.

Reading the Referee's comments, we realized that the presentation of the previous manuscript was not in the best form. Also, the Referee made a great point for our zero-temperature mean-field calculations where the IC-CDW can survive. Thus, we have newly performed numerical calculations, employing one of the well-known numerical approximation methods to incorporate incommensurate CDW orders [R20]. Spending significant amounts of effort and resources on the numerical calculations, we obtain a better picture of the clean monolayer kagome metals at zero temperature. Notably, the intriguing competing physics between IC-CDW and 2×2 CDW is uncovered, as summarized in Fig.R11. Our new results can be considered as a demonstration of the new physics originating from Type-II VHS points (P_i) that are off high-symmetry momenta (M_i) at zero temperature.

Let us briefly sketch how our mean-field calculations at zero temperature are extended to incorporate IC-CDW orders by extending the original ansatz for 2×2 CDW orders only. The possible Q vectors of IC-CDW orders associated with (AB, BC, CA) bonds are constructed by connecting the rearranged VHS points,

$$\mathbf{Q}_{i,j}^{IC} = (\mathbf{P}_i^{(2)} - \mathbf{M}_1, \mathbf{P}_j^{(3)} - \mathbf{P}_i^{(2)}, \mathbf{M}_1 - \mathbf{P}_j^{(3)}), \quad (\text{R5})$$

where $P_i^{(n)}$ is a shifted Type-II VHS point from M_n by $d\mathbf{q}_i$ (See Fig. R12 i,ii of (a-d)). Here, $d\mathbf{q}_{1,3} = \mp[0.0498\mathbf{b}_1 - 0.0315\mathbf{b}_2]$, $d\mathbf{q}_{2,4} = \mp[0.315\mathbf{b}_1 - 0.0498\mathbf{b}_2]$ are defined with a 1×1 reciprocal vector $\mathbf{b}_{1,2}$. Depending on choosing different type-II VHSs, ($P_i^{(2)}, P_j^{(3)}$), there are 16 number of cases of

IC-CDWs, distinguished by deviated Q vectors, $dQ_{i,j}^{IC}$, from those of 2×2 CDW orders $Q_{2 \times 2} \equiv (M_2 - M_1, M_3 - M_2, M_1 - M_3)$, whose explicit form is given by

$$dQ_{i,j}^{IC} \equiv Q_{i,j}^{IC} - Q_{2 \times 2} = (dq_i, dq_j - dq_i, -dq_j). \quad (\text{R6})$$

Here, the D_{2h} symmetry simplifies the possible cases into four ($i = 3, j \in 1, 2, 3, 4$) and the associated possible $3Q$ and $1Q$ IC-CDW orders are taken into account (See Fig.R12). Note that by $3Q$ orders we mean that the relative amplitudes of (Q_{AB}, Q_{BC}, Q_{CA}) bonds are equivalent, while $1Q$ CDW order has only one finite amplitude of bond order among (Q_{AB}, Q_{BC}, Q_{CA}) bonds.

Technically, we have performed the supercell calculations with commensurate approximations, which is one of the most popular and well-established theoretical methods used for IC-CDW analysis. Following the previous literature [R20], we were able to approximate the Q vector of IC-CDW to the closest rational numbers of the 1×1 reciprocal vector, \mathbf{b}_i , which reproduce similar physical properties of the quasicrystal. For example, within 20×32 supercell approximation, IC-CDW wave vectors are approximated as

$$dq_{1,3} = \mp[0.0498\mathbf{b}_1 - 0.0315\mathbf{b}_2] \rightarrow dq_{1,3}^{\text{app}} \mp \left[\frac{1}{20}\mathbf{b}_1 - \frac{1}{32}\mathbf{b}_2 \right], \quad (\text{R7})$$

$$dq_{2,4} = \mp[0.0314\mathbf{b}_1 - 0.0498\mathbf{b}_2] \rightarrow dq_{2,4}^{\text{app}} \mp \left[\frac{1}{32}\mathbf{b}_1 - \frac{1}{20}\mathbf{b}_2 \right]. \quad (\text{R8})$$

The accuracy of these approximations increases as reducing the difference between actual values (dq_i) and approximated ones (dq_i^{app}) by enlarging the supercell size (say, $N_1 \times N_2$), as shown in Fig.R13, where approximation error, $(|dq_i - dq_i^{\text{app}}|/dq_i)$, is plotted as a function of the supercell size, (N_1, N_2) . We note that our approximated IC-CDW Hamiltonian is already a 1920×1920 dimension, and increasing the matrix size is forbiddingly demanding since both the time and memory cost scale grows quadratically, $\propto (N_1 N_2)^2$ [R21].

Based on our 20×32 supercell calculations, we have obtained the new phase diagram at $U \neq 0, V = 0$, as depicted in Fig.R11. As the Referee expected, IC-CDW orders clearly appear ($U > -0.164$). However, the 2×2 CDW orders, such as SD-2 and ISD-1, still survive in the broad range ($U < -0.164$). We highlight that the competition between IC-CDW and 2×2 CDW phases is very versatile under varying interaction strength due to the tiny energy differences (i.e. $|E_{IC}^{\text{min}} - E_{2 \times 2}^{\text{min}}|/E_{IC}^{\text{min}} = 2 \times 10^{-5}$ at $U = 0.2, V = 0$), and more sophisticated analysis with advanced numerical methods is required for investigating these competing physics.

Before closing the discussion about the IC-CDW, we would like to point out intriguing future

research works associated with the IC-CDW order parameters in the monolayer systems. First, the quasi-long range properties of the IC-CDW order parameters indicate the possibility of the topological phase transitions such as the Kosterlitz-Thouless transitions. Though an experimental identification of the topological phase transitions is challenging, it would still be an intriguing theoretical problem. Second, the interplay between the IC-CDW and the dimensionality with the van Hove singularities of the layered kagome metals is also an interesting problem. It is well known that a quasi-long range order parameter of a monolayer system can become a long-range order with stronger interactions or layered systems. Interestingly, the competition between 2×2 CDW and IC-CDW is also recently observed in the hole-doped CsV_3Sb_5 bulk samples, which is attributed to the symmetry breaking [R22]. Both works are exciting problems but are beyond the current work's scope. In this respect, we believe that our revised manuscript offers timely opportunities to encounter new physics in the kagome metal community, calling for future sophisticated theoretical and experimental studies. In the revised manuscript, we have added a new paragraph to discuss possible intriguing physics originating from IC-CDW orders in the "Discussion and Conclusion" section in the main text (Page 13, xiii) and expanded a new section in Supplementary Note 10.

FIG. R11. **Exemplary mean-field phase diagram of monolayer AV_3Sb_5 including incommensurate charge density wave (IC-CDW) orders.** We considered IC-CDWs introduced in Fig.R12. The phase diagram is calculated at the type-II VHS filling $\mu = -6$ meV in U space for $V = 0$. While the IC-CDW orders are energetically favored over the 2×2 CDW orders in the region ($U > -0.164$), the 2×2 CDW orders are intact and survive in the region ($U < -0.164$).

FIG. R12. **Schematic illustrations of incommensurate CDW (IC-CDW) order parameters.** (i) and (ii)-the columns show the wave-vectors of the CDW orders in the 1×1 and $\sqrt{3}\times 1$ Brillouin zones (BZs), respectively. (iii)-th column delineates the corresponding 3Q bond modulations in real space to exemplify the case. **(a-d)** IC-CDW orders. **(e)** Commensurate 2×2 CDW order. The Type-I (Type-II) VHSs are marked by blue (red) circles in the 1×1 and $\sqrt{3}\times 1$ BZs (i,ii of **a-e**). The order parameters are constructed, such that one of the three Q vectors, Q_{BC} , connects two (equivalent or inequivalent) type-II P_i out of four (P_1, P_2, P_3 , and P_4) and the rest two Q vectors, (Q_{AB}, Q_{AC}), connect the chosen P_i and the VHS at Γ , respectively. This construction results in four symmetry-inequivalent order parameters listed in **a-d**. The shifted bond Q vectors $d\mathbf{Q}$ are chosen as $d\mathbf{Q}_{3,3}^{IC}, d\mathbf{Q}_{3,4}^{IC}, d\mathbf{Q}_{3,1}^{IC}$, and $d\mathbf{Q}_{3,2}^{IC}$ (i,ii of **a-d**). The 2×2 CDW corresponds to $d\mathbf{Q} = 0$ (i,ii of **e**).

FIG. R13. **The error of commensurate approximation depending on the supercell size $N_1, N_2 < 100$.** To employ commensurate approximations, we approximate the two irrational values (0.0498, 0.0315) to the closest rational numbers, expressed as a fraction n/N of two integers (n, N). Here, the numbers are inherited from $d\mathbf{q}_i$ (See Eqs.R7,R8), and the error is defined as $|\text{Exact} - \text{Approximated value}|/\text{Exact value}$. The approximated values (1/20, 1/32) give the local minimum in the error with the actual values (0.0498, 0.0315), thus, we perform 20×32 supercell calculations.

R: (7) It seems that the authors simply restricted the mean-field to the 2x2 CDW states studied extensively for the D6h kagome bulk material. This limits the ability to reveal the new physics of the monolayer with a different symmetry, and especially the intrinsic symmetry breaking effects due to the $\sqrt{3}\times 1$ structure.

A: We believe that this comment is answered above in detail.

R: Moreover, how is such an approach justified? Doesn't the DFT already take into account at least part of the Coulomb interaction?

A: We thank the Referee for asking these questions. Our approach to understanding symmetry-broken phases is the standard mean-field analysis of interacting electronic systems [R23–R25]. It is well-known that the Coulomb interaction effects of many electrons are extremely difficult to capture precisely. Even for DFT calculations, it is well-established that the double counting problems are notorious [R26, R27]. Thus, some types of approximation must be employed unless exactly solvable systems are considered.

Our mean-field analysis is a standard alternative to the above-mentioned shortcomings of the DFT calculation, which is to start with a reasonable tight-binding model constructed from the DFT calculation and add the effective Coulomb interactions. For the given tight-binding model, the effective Coulomb interactions must be thought of as a function of the original Coulomb interaction of all electrons and its screening from the additional bands excluded from the tight-binding model. Evaluating the effective Coulomb interactions is difficult, so in practice, one treats the effective Coulomb interactions as parameters, which allows for the investigation of available symmetry-broken phases nicely.

We would like to mention that the DFT and the mean-field analysis are complementary to each other, as demonstrated in many works in strongly correlated systems including cuprates, pnictides, and even bulk kagome metals [R4, R12–R14, R23, R24, R28].

R: (8) The authors found TRSB flux phases when the chemical potential is moved close to the type-I VHS, which derives from the two VHS points at M_1 least affected by the $\sqrt{3}\times 1$ folding. This is interesting.

A: We thank the Referee for considering our finding interesting.

R: The authors calculated the Chern numbers associated with each band in Fig. 4. This is the only figure that offered some details of the band structure in the symmetry-breaking phases of the monolayer.

A: The Referee appears concerned about the details of the band structure in the symmetry-broken phases of the monolayer. To address the concern, we included the band structure of the other symmetry-breaking phases in Supplementary Fig. S10, including the ISD-1 and SD-2 CDWs of AV_3Sb_5 for $A = K, Rb, Cs$, as shown in Fig. R14 below:

FIG. R14. The band structures of symmetry-breaking phases in the monolayer AV_3Sb_5 .

R: They also calculated the anomalous Hall conductivity. There is a similar study on the pristine kagome lattice by Zhou and Wang arXiv:2110.06266. The TRSB current patterns look very similar. Are there magnetic fluxes threading the rest of the triangles? The authors should discuss the physics in more detail and compare to the case studied for the pristine kagome lattice.

A: We thank the Referee for pointing to the fine study by Zhou and Wang. As the Referee correctly noted, the complex CDW introduced in arXiv:2110.06266 (Ref. [R29]) can be considered as a bulk counterpart of the TRSB-1 and TRSB-2 CDWs that we studied in the monolayer. As compared in Fig. R15, the current patterns of our TRSB-1 and TRSB-2 are the same as those of the complex CDW order of Ref. [R29] with the magnetic fluxes threading in all the hexagons and triangles. Notably, the Chern Fermi pockets are featured in our system as well, realizing the phase referred to as a doped orbital Chern insulator by Zhou and Wang. We agree that we missed these essential physics in our discussion. To make up for it, we included the following words in the revised manuscript (Page 12, xii):

“... As shown in Fig. 5a, the different Chern numbers between the two phases arise due to the concurrent sign-change of the Berry curvature at high-symmetry momenta M_1 , K_1 , and K_2 . We note that the monolayer in the time-reversal symmetry broken CDW phase hosts the Fermi pockets that carry the Berry curvature, referred to as the Fermi Chern pockets [48] (see Supplementary Fig. S11). ...”

R: Fig. 4a indicates different Fermi pockets at M_1 and M_2 , which are different from Zhou and Wang and presumably come from D_{6h} symmetry breaking. The authors may be able to provide some experimental signatures for distinguishing the bulk and monolayer materials.

A: This is a brilliant suggestion! Despite the similarity of the order parameters, they give rise to distinct physical manifestations, suggesting an experimental indicator to discern the bulk and monolayer systems. We agree with the Referee and sincerely thank him/her for making such an important observation.

As suggested, we explicitly show that the Chern numbers are distributed differently over the bands near the Fermi level between the D_{2h} monolayer and the D_{6h} bulk, as delineated in Fig. R15. In particular, the different distributions of the Berry curvatures on the Fermi surface are found near the M_1 and M_2 (as well as K_1 and K_2) points, which can be mainly attributed to distinct parameters that are chosen such that they depict the absence of the C_{3z} rotational symmetry of the monolayer. In detail, the threading flux values of the TRSB phase in Ref. [R29] are different from our TRSB-1 and TRSB-2 phases (Table. R2). The complex CDW phase in Ref. [R29] has flux values as $\phi_{1,\dots,4} = (0.840, -0.113, -0.291, 0.014)\pi$ while our TRSB-1 and TRSB-2 phases have the flux values as $\phi_{1,\dots,4} = (0.318, -0.103, -0.159, 0.051)\pi$ and

$\phi_{1,\dots,4} = (0.313, -0.108, -0.157, 0.054)\pi$, respectively.

Correspondingly, distinct anomalous Hall conductivities were calculated between the bulk and monolayer systems. The bulk system has the anomalous Hall conductivity of around $-1.1 \frac{e^2}{h}$, whereas the monolayer in TRSB-1 and TRSB-2 phases lead to -0.26 and $-0.06 \frac{e^2}{h}$, respectively. Here, e is electronic charge and h is the Plank constant. This difference in the anomalous Hall conductivities should be crucial for distinguishing the bulk and monolayer systems. Thus, the following mention is included in the revised manuscript (Page 12, xii):

“... However, unlike the bulk case [48], asymmetric distribution of the Berry curvature occurs near the M_1 and M_2 in the monolayer due to the symmetry lowering. This results in distinct experimental observables, such as anomalous Hall conductivity σ_{xy} (see detailed comparison between bulk and monolayer in Supplementary Note 5). ...”

Moreover, we revised Fig. 4 to show all the fluxes threading triangles. During the revision, we found that the directions of threading fluxes used to obtain data in the previous manuscript were opposite to those of current data in Fig. R15, which means that all the signs of topological quantities such as the Berry curvature, Chern number, and anomalous Hall conductivity can be defined with an opposite sign. To keep consistency, we thus fix the sign issue in Fig. 5 in the revised version.

Model	ϕ_1	ϕ_2	ϕ_3	ϕ_4	t_1	t_2	t_3
[R29]	0.840π	-0.113π	-0.291π	0.014π	$0.345 \pm 0.162i$	$0.499 \pm 0.123i$	$0.349 \pm 0.110i$
TRSB-1	0.318π	-0.103π	-0.159π	0.051π	$0.42 \pm 0.07i$	$0.41 \pm 0.07i$	$0.43 \pm 0.07i$
TRSB-2	0.313π	-0.108π	$\pm 0.157\pi$	0.054π	$0.42 \pm 0.07i$	$0.41 \pm 0.07i$	$0.43 \pm 0.07i$

TABLE R2. **Magnetic fluxes and complex hopping parameters of the TRSB phases in Fig. R15.** The unit of hopping parameters is the electron volt (eV). The \pm sign of the imaginary hopping constants denotes the direction of the bond current depicted in Fig R15.

-
- [R1] Cahangirov, S., Topsakal, M., Aktürk, E., Şahin, H. & Ciraci, S. Two- and one-dimensional honeycomb structures of silicon and germanium. *Phys. Rev. Lett.* **102**, 236804 (2009).
- [R2] Şahin, H. *et al.* Monolayer honeycomb structures of group-iv elements and iii-v binary compounds: First-principles calculations. *Phys. Rev. B* **80**, 155453 (2009).

FIG. R15. **Current pattern, band structure, Fermi surface, and Berry curvature of various TRSB phases.** **a** Current pattern, band structure, Fermi surface, and Berry curvature of the TRSB phase studied in Zhou and Wang [R29]. **b,c** Current pattern, band structure, Fermi surface, and Berry curvature of the TRSB-1 and TRSB-2 phases were studied in this work. The magnitudes of threading fluxes $\phi_1, \phi_2, \phi_3,$ and ϕ_4 as well as the hopping parameters are given in Table R2. The different hopping parameters are displayed in different colors. Chern numbers of each band are displayed in the band structures. Energy contours and Berry curvature distribution of the band lying at the Fermi level (colored in red in the band structures) are drawn in the bottom panel.

- [R3] Shao, Y., Shao, M., Kawazoe, Y., Shi, X. & Pan, H. Exploring new two-dimensional monolayers: pentagonal transition metal borides/carbides (penta-tmb/cs). *J. Mater. Chem. A* **6**, 10226–10232 (2018).
- [R4] Tan, H., Liu, Y., Wang, Z. & Yan, B. Charge density waves and electronic properties of superconducting kagome metals. *Phys. Rev. Lett.* **127**, 046401 (2021).
- [R5] Waldecker, L. *et al.* Rigid band shifts in two-dimensional semiconductors through external dielectric screening. *Phys. Rev. Lett.* **123**, 206403 (2019).
- [R6] Kim, M. *et al.* Control of electron-electron interaction in graphene by proximity screening. *Nature communications* **11**, 1–6 (2020).
- [R7] Roy, B. & Foster, M. S. Quantum multicriticality near the dirac-semimetal to band-insulator critical point in two dimensions: A controlled ascent from one dimension. *Phys. Rev. X* **8**, 011049 (2018).

- [R8] Aryasetiawan, F. *et al.* Frequency-dependent local interactions and low-energy effective models from electronic structure calculations. *Phys. Rev. B* **70**, 195104 (2004).
- [R9] Li, H. *et al.* Observation of unconventional charge density wave without acoustic phonon anomaly in kagome superconductors AV_3Sb_5 ($A = Rb, Cs$). *Phys. Rev. X* **11**, 031050 (2021).
- [R10] Liu, G. *et al.* Observation of anomalous amplitude modes in the kagome metal CsV_3Sb_5 . *Nature communications* **13**, 1–8 (2022).
- [R11] LaBollita, H. & Botana, A. S. Tuning the van hove singularities in AV_3Sb_5 ($A = K, Rb, Cs$) via pressure and doping. *Phys. Rev. B* **104**, 205129 (2021).
- [R12] Wu, X. *et al.* Nature of unconventional pairing in the kagome superconductors AV_3Sb_5 ($A = K, Rb, Cs$). *Phys. Rev. Lett.* **127**, 177001 (2021).
- [R13] Jeong, M. Y. *et al.* Crucial role of out-of-plane Sb p orbitals in van hove singularity formation and electronic correlations in the superconducting kagome metal CsV_3Sb_5 . *Phys. Rev. B* **105**, 235145 (2022).
- [R14] Denner, M. M., Thomale, R. & Neupert, T. Analysis of charge order in the kagome metal AV_3Sb_5 ($A = K, Rb, Cs$). *Phys. Rev. Lett.* **127**, 217601 (2021).
- [R15] Lee, J. M. *et al.* Stable flatbands, topology, and superconductivity of magic honeycomb networks. *Phys. Rev. Lett.* **124**, 137002 (2020).
- [R16] Mermin, N. D. & Wagner, H. Absence of ferromagnetism or antiferromagnetism in one- or two-dimensional isotropic heisenberg models. *Phys. Rev. Lett.* **17**, 1133–1136 (1966).
- [R17] Hohenberg, P. C. Existence of long-range order in one and two dimensions. *Phys. Rev.* **158**, 383–386 (1967).
- [R18] Coleman, S. There are no goldstone bosons in two dimensions. *Communications in Mathematical Physics* **31**, 259–264 (1973).
- [R19] Nie, L., Tarjus, G. & Kivelson, S. A. Quenched disorder and vestigial nematicity in the pseudogap regime of the cuprates. *Proceedings of the National Academy of Sciences* **111**, 7980–7985 (2014).
- [R20] Yu, G., Wu, Z., Zhan, Z., Katsnelson, M. I. & Yuan, S. Dodecagonal bilayer graphene quasicrystal and its approximants. *npj Computational Materials* **5**, 1–10 (2019).
- [R21] Troyer, M. & Zürich, E. Classical and quantum monte carlo algorithms and exact diagonalization (2004).
- [R22] Kautzsch, L. *et al.* Incommensurate charge-stripe correlations in the kagome superconductor $CsV_3Sb_{5-x}Sn_x$. *arXiv:2207.10608 [cond-mat]* (2022).
- [R23] Sau, J. D. & Sachdev, S. Mean-field theory of competing orders in metals with antiferromagnetic exchange interactions. *Phys. Rev. B* **89**, 075129 (2014).
- [R24] Fernandes, R. M. & Schmalian, J. Competing order and nature of the pairing state in the iron pnictides. *Phys. Rev. B* **82**, 014521 (2010).
- [R25] Wen, X. G. Mean-field theory of spin-liquid states with finite energy gap and topological orders. *Phys. Rev. B* **44**, 2664–2672 (1991).

- [R26] Anisimov, V. I., Aryasetiawan, F. & Lichtenstein, A. I. First-principles calculations of the electronic structure and spectra of strongly correlated systems: the LDA + U method. *Journal of Physics: Condensed Matter* **9**, 767–808 (1997).
- [R27] Ylvisaker, E. R., Pickett, W. E. & Koepnik, K. Anisotropy and magnetism in the LSDA + U method. *Phys. Rev. B* **79**, 035103 (2009).
- [R28] Aichhorn, M., Pourovskii, L. & Georges, A. Importance of electronic correlations for structural and magnetic properties of the iron pnictide superconductor LaFeAsO. *Phys. Rev. B* **84**, 054529 (2011).
- [R29] Zhou, S. & Wang, Z. Chern fermi-pockets and chiral topological pair density waves in kagome superconductors. *arXiv:2110.06266 [cond-mat]* (2021).

REVIEWER COMMENTS

Reviewer #1 (Remarks to the Author):

The Authors in the present manuscript has carried out all the analysis corresponding to the previous questions. The answers look okay to me and the present manuscript is eligible for the publication in nature communication journal.

Reviewer #2 (Remarks to the Author):

I think that all previously raised issues have been satisfactorily addressed, and the manuscript has been improved greatly. I can therefore recommend the current version to Nature Communications.

Reviewer #3 (Remarks to the Author):

The authors responded to all of my comments, as well as those of the other reviewers, with extensive and detailed calculations. These new results do have the potential to establish a physical picture and improve the manuscript. Since this is the second round of review, I would just focus on my main concerns affecting the main result and conclusion of the paper over which similar questions and comments had been raised by all the reviewers.

(1) The authors have now calculated the phonon spectrum of the monolayer kagome metal in the D2h structure. The structure is unstable as shown in Fig. 3e, and “negative” phonon energies appear for a broad range of momenta in the zone. This is in contrast to the bulk kagome metals in the D6h structure where the unstable phonon modes are localized around the M points. It further substantiates my concern (and other reviews too) about whether the 2x2 reconstruction established for bulk D6h kagome metals is the leading instability in this theoretically proposed monolayer structure.

This is a major issue because if the authors consider only the commensurate 2x2 reconstructions, the phonon spectrum of the reconstructed structure looks still unstable. In Fig. 6b calculated for the ISD-1, one phonon branch near the zone center Gamma point is still unstable, exhibiting a highly unusual dispersion where acoustic phonons in a stable structure should be expected. This may signify that the true instability is toward the incommensurate CDW. The effective model mean-field calculations given in the supplemental Fig.S19 and associated discussions indeed show that the commensurate 2x2 CDW is replaced by incommensurate CDW for repulsive interactions. It is inaccurate to conclude that the 2x2 commensurate ISD phase is stable in the monolayer D2h kagome metals.

(2) Throughout the response and in several places in the revised manuscript, the authors now mention the incommensurate CDW, but downplays its significance by invoking the Mermin-Wagner theorem. This is unsuitable. The calculations in the manuscript are done at zero temperature and address the possible quantum states. It is understood that in the presence of gapless excitations in 2D, such as in an incommensurate CDW, the long-range ordered states at zero temperature exhibit quasi-long range order with power-law correlations and Kosterlitz-Thouless type of transitions at finite temperatures and they are physically well defined and observable. It is thus inaccurate to conclude that at any non-zero temperature, the IC-CDW phases completely disappear due to the Mermin-Wagner theorem. If we follow the authors' argument, the formation of the 2D monolayer crystal is where continuous translation symmetry is broken, and the theorem at the face value would have said 2D crystals are prohibited at any finite temperatures.

Because of these issues, the results, discussions, and conclusions in the revised version of the manuscript are not solid and at times misleading. They need to be addressed before further consideration.

Summary of changes

- In the main text and Supplementary Note 5.1, we have extended the imaginary phonon mode analysis to confirm the 2×2 CDW structure is the leading instability.
- In the main text, we have replaced the phonon bands of the ISD-1 phase in Fig. 6b with the newly obtained data, which is clean of artificial negative phonon modes.
- In Supplementary Note 5.2, we have discussed the correction of artificial negative phonon modes of the ISD-1 phase.
- In the main text, we have emphasized the significance of the IC-CDW in the monolayer kagome metals and added a few words about the issues of the quasi-long range IC-CDW order in the “Discussion and Conclusion” section.
- In Supplementary Note 11, we have replaced the mean-field phase diagrams at two different chemical potentials, Fig.S21, to apparently show the filling dependence of the competing behaviors of incommensurate CDW.
- The modified parts of the main text are colored red (blue) font in the response letter (revised main text).

Answers to the 1st Referee's comments

Referee (R) : The Authors in the present manuscript has carried out all the analysis corresponding to the previous questions. The answers look okay to me and the present manuscript is eligible for the publication in nature communication journal.

Authors (A) : We thank the Referee for the time and effort. We are also grateful for the positive assessment and the recommendation for publication.

Answers to the 2nd Referee's comments

R: I think that all previously raised issues have been satisfactorily addressed, and the manuscript has been improved greatly. I can therefore recommend the current version to Nature Communications.

A: We thank the Referee for reviewing our revised manuscript. We are happy to receive the positive assessment and the recommendation for publication in Nature communications.

Answers to the 3rd Referee's comments

R: The authors responded to all of my comments, as well as those of the other reviewers, with extensive and detailed calculations. These new results do have the potential to establish a physical picture and improve the manuscript.

A: We thank the Referee for the time and effort for the second review. We are happy to find the Referee's positive assessment.

R: Since this is the second round of review, I would just focus on my main concerns affecting the main result and conclusion of the paper over which similar questions and comments had been raised by all the reviewers.

A: We thank the Referee for the comments. Below, we addressed all the concerns raised by the Referee in detail.

R: (1) The authors have now calculated the phonon spectrum of the monolayer kagome metal in the D2h structure. The structure is unstable as shown in Fig. 3e, and “negative” phonon energies appear for a broad range of momenta in the zone. This is in contrast to the bulk kagome metals in the D6h structure where the unstable phonon modes are localized around the M points. It further substantiates my concern (and other reviews too) about whether the 2x2 reconstruction established for bulk D6h kagome metals is the leading instability in this theoretically proposed monolayer structure.

A: We thank the Referee for sharing the insight on the phonon energy spectrum. As the Referee correctly pointed out, the phonon bands exhibit a broad-ranged region of negative energies (reproduced in Fig. R1a), which made the Referee concerned about the leading instability of the monolayer. To dispel the concern, we have newly performed total energy calculations and found additional evidence that the 2×2 CDW reconstruction is the primary instability.

In Fig. R1b, the total energy profiles with different CDW reconstructions are presented. We choose the high symmetry momenta X and M in addition to $q_1 = \frac{1}{6}\mathbf{b}'_1$ and $q_2 = \frac{1}{10}\mathbf{b}'_2$ of the negative energy phonon modes. Here, we note that $\mathbf{b}'_1, \mathbf{b}'_2$ are reciprocal vectors of $\sqrt{3} \times 1$ unit cells, and q_1, q_2 are the closest commensurate wave vectors to the nesting vectors of the type-II VHS points for the incommensurate CDWs.

The minimum of the total energy appears with the momentum M , which corresponds to the 2×2 CDW reconstruction. We also note that the magnitude of the condensation energy with the momentum M (~ -9.7 meV/f.u.) is much lower than the one with the other momenta (~ -3.5 meV/f.u.). We also explore the reconstructions with a few more commensurate momenta and obtain similar results. Thus, we conclude that the leading instability in the monolayer is the 2×2 CDW reconstruction.

Thanks to the Referee’s insight, we were able to improve our study, by considering the possibility of other instabilities featured in the broadness of the softened phonon mode. We added these results in the main text on Page 7, vii (changes are colored in red):

“... We also find that these 2×2 CDW instabilities are generically present in monolayer AV_3Sb_5 for $A = K, Rb,$ and Cs (see Supplementary Fig. S3). We note that the monolayer hosts the broadened softened phonon modes in the momentum space compared to the bulk [17], which signals various CDW instabilities other than 2×2. Nonetheless, we confirm that the 2×2 CDW instability is a leading instability by performing energy profile analysis for the various negative phonon modes (see Supplementary Note 5.1). Thus, it is imperative to consider the 2×2 3Q CDWs ...”

FIG. R1. **Negative phonon mode analysis of the AV_3Sb_5 monolayer.** **a** Phonon bands of the pristine KV_3Sb_5 monolayer at the smearing factor $\sigma = 0.1$ eV. Here, $q_1 = \frac{1}{6}\mathbf{b}'_1$ and $q_2 = \frac{1}{10}\mathbf{b}'_2$ (\mathbf{b}'_1 and \mathbf{b}'_2 are the reciprocal lattice vectors of the $\sqrt{3} \times 1$ unit cell) are the closest commensurate vectors to the incommensurate nesting vectors mediating the type-II VHS points (see also Fig.3a in the main text). **b** Energy profiles (per formula unit) as a function of the amplitude of the phonon mode for various negative energy states. The corresponding energy and momenta are indicated in **a**. We stress that the minimum of total energy appears at M (~ -9.7 meV/f.u.), which corresponds to the 2×2 CDW reconstruction.

R: This is a major issue because if the authors consider only the commensurate 2×2 reconstructions, the phonon spectrum of the reconstructed structure looks still unstable. In Fig. 6b calculated for the ISD-1, one phonon branch near the zone center Gamma point is still unstable, exhibiting a highly unusual dispersion where acoustic phonons in a stable structure should be expected. This may signify that the true instability is toward the incommensurate CDW.

A: This is a keen observation. The phonon spectrum of the 2×2 reconstruction has small negative energies near Γ , as pointed out by the Referee. Extensive additional calculations are performed to check the issue, by increasing the supercell size from 4×4 to 8×8 . The more accurate and refined phonon bands are obtained and presented in Fig. R2. Near the Γ point, the supercell size dependence is illustrated in (i) and (ii). As expected, the 8×8 supercell calculation gives the stable phonon spectrum.

We, therefore, conclude that the 2×2 CDW reconstruction is free of the stability issue from the phonon spectrum, and the seemingly unstable phonon spectrum in the previous manuscript is from computational accuracy.

In the revised version, we replaced Fig. 6b with the more accurate new results (colored in blue in Fig. R2), which is clean of the artifact. In addition, we have mentioned the above discussions in Supplementary Note 5.2.

FIG. R2. **Convergence test of the negative phonon energies of the ISD-1 structure.** The phonon bands of the ISD-1 structure are calculated using three different supercells (SCs): 4×4 , 6×6 , and 8×8 . The erroneous negative energies, which appeared in the previous 4×4 SC calculations, converge to positive ones as we increase the computational accuracy using a large SC. The insets (i) and (ii) display the magnified views of the boxed regions near Γ . The arrows indicate the converging behavior of the negative branches (from 4×4 to 8×8 SC). The left panel depicts the Brillouin zone (solid line) and the corresponding high-symmetry points of the 2×2 unit cell.

R: The effective model mean-field calculations given in the supplemental Fig.S19 and associated discussions indeed show that the commensurate 2×2 CDW is replaced by incommensurate CDW for repulsive interactions. It is inaccurate to conclude that the 2×2 commensurate ISD phase is stable in the monolayer D2h kagome metals.

A: We thank the referee for the comment. As pointed out by the Referee, the incommensurate CDW replaces the 2×2 CDW in some portions of the provided phase diagrams.

Yet, we would like to stress that the 2×2 CDW is stable even with introducing the incommensurate CDW order parameters. To demonstrate this, we additionally calculated the phase diagrams at different fillings $\mu = -7$ meV and $\mu = -5$ meV in Fig. R3. The stark difference between two phase diagrams has been shown at repulsive interaction ($U > 0$). The commensurate CDW (ISD-2) dominates over the incommensurate CDW for $\mu = -7$ meV while the incommensurate CDW is dominant for $\mu = -5$ meV.

We emphasize that the small difference of the chemical potentials ($\delta\mu = 2$ meV) triggers the qualitative differences between (a) and (b) of Fig. R3. Our calculations indicate that the monolayer kagome metal systems show intriguing and delicate competition physics between the commensurate and incommensurate CDWs, which deserves future refined research.

In the revised manuscript, we added a few words in the main text (Page 13, xiii) and replaced Fig.S21 to apparently show the filling dependence of the competing behaviors of IC- CDW.

FIG. R3. Chemical potential dependence of incommensurate CDW phase. Phase diagrams are calculated at two different fillings (a) $\mu = -5$ meV and (b) $\mu = -7$ meV with (right) and without (left) the incommensurate CDW order. These are representative fillings near the type-II filling ($\mu = -6$ meV), which lead to a contrasting impact on the competition between the commensurate and incommensurate CDWs. For repulsive interactions ($U > 0$), the commensurate CDW (ISD-2) dominates over the incommensurate CDW for $\mu = -7$ meV, whereas the incommensurate CDW is dominant for $\mu = -5$ meV. The convergent results are obtained by using 20×32 supercell calculations with 61×61 \mathbf{k} mesh points in the reduced Brillouin zone.

R: (2) Throughout the response and in several places in the revised manuscript, the authors now mention the incommensurate CDW, but downplays its significance by invoking the Mermin-Wagner theorem. This is unsuitable. The calculations in the manuscript are done at zero temperature and address the possible quantum states. It is understood that in the presence of gapless excitations in 2D, such as in an incommensurate CDW, the long-range ordered states at zero temperature exhibit quasi-long range order with power-law correlations and Kosterlitz-Thouless type of transitions at finite temperatures and they are physically well defined and observable. It is thus inaccurate to conclude that at any non-zero temperature, the IC-CDW phases completely disappear due to the Mermin-Wagner theorem. If we follow the authors' argument, the formation of the 2D monolayer crystal is where continuous translation symmetry is broken, and the theorem at the face value would have said 2D crystals are prohibited at any finite temperatures.

A: We would like to thank the Referee for the comments and questions about the incommensurate CDW (IC-CDW). Incorporating the discussion about the IC-CDW, we believe our work has been much improved and even may open up new exciting future research activities of monolayer kagome metals.

We emphasize that the validity of our calculations for the mean-field phase diagrams has been thoroughly checked at low temperatures, where the temperature is smaller than all other energy scales of AV_3Sb_5 , by performing extensive calculations. Based on the Mermin-Wagner-Hohenberg (MWH) theorem, our phase diagrams are valid at low temperatures below the Kosterlitz-Thouless (KT) transition temperature associated with the IC-CDW, if the KT phase of the IC-CDW exists. In fact, we have already mentioned the possibility of the KT transition associated with the quasi-long range order of the IC-CDW by stating that "First, the quasi-long range properties of the IC-CDW order parameters indicate the possibility of the topological phase transitions such as the Kosterlitz-Thouless transitions." in the previous response (Answer (6) of the 3rd Referee). Thus, we believe that our conclusions based on the calculations are concrete and applicable to the 2D monolayer systems at low temperatures.

The presence or absence of the quasi-long range IC-CDW orders is certainly an important issue. Also, the validity of the Mermin-Wagner theorem in the monolayer AV_3Sb_5 is also an intriguing research topic, which has been known to have loopholes with such as long-range interactions, anisotropic couplings, finite-size effects, and disorder [R1, R2]. Yet, we would like to point out that our focus of the current work is to provide the first global picture of the monolayer AV_3Sb_5 systems in terms of the mean-field phase diagram. Thus, we believe the issues of the quasi-long

range IC-CDW orders are beyond the scope of the current work and leave exciting problems for future works.

In the revised manuscript, we have added more words about the issues of the quasi-long-range IC-CDW orders in the “Discussion and Conclusion” section in the main text on Page 13, xiii (changes are colored in red):

“... Thus, not only the electron filling but also interaction strengths are important to stabilize the IC-CDW phases. At any non-zero temperature, the quasi-long range IC-CDW order can exist, and uncovering the possibility of topological phase transitions such as Kostelitz-Thouless transitions is certainly an important issue in both theoretical and experimental regards. ...”

R: Because of these issues, the results, discussions, and conclusions in the revised version of the manuscript are not solid and at times misleading. They need to be addressed before further consideration.

A: We believe that our revised manuscript is improved significantly. In particular, the logical flow of the revised manuscript has been strengthened by preparing the response to the Referee’s comments and questions. Thus we believe our work is suitable for publication in Nature Communications, and we hope the Referee agrees with us.

[R1] Halperin, B. I. On the hohenberg–mermin–wagner theorem and its limitations. *Journal of Statistical Physics* **175**, 521–529 (2019).

[R2] Palle, G. & Sunko, D. K. Physical limitations of the hohenberg–mermin–wagner theorem. *Journal of Physics A: Mathematical and Theoretical* **54**, 315001 (2021).

REVIEWERS' COMMENTS

Reviewer #3 (Remarks to the Author):

Third Referee Report on "Monolayer Kagome Metals AV₃Sb₅" by S.-W. Kim et al.

The authors responded to my concerns by improving their calculations using larger supercells that removed the ambiguity in the numerical data in the previous version. They have also reworded the discussions with regard to the usage of Mermin-Wagner theorem and the incommensurate charge density waves. I think the revised manuscript has been significantly improved and technically sound. I can now recommend publication.

Answers to the 3rd Referee's comments

Referee (R) : The authors responded to my concerns by improving their calculations using larger supercells that removed the ambiguity in the numerical data in the previous version. They have also reworded the discussions with regard to the usage of Mermin-Wagner theorem and the incommensurate charge density waves. I think the revised manuscript has been significantly improved and technically sound. I can now recommend publication.

Authors (A) : We thank the Referee for the time and effort. We are also grateful for the recommendation for publication.